# Complexity Matters: Rethinking the Latent Space for Generative Modeling

**Tianyang Hu[1], Fei Chen[1], Haonan Wang[2], Jiawei Li[1],
Wenjia Wang[3], Jiacheng Sun[1,*], Zhenguo Li[1]**

[1] Huawei Noah's Ark Lab, [2] National University of Singapore
[3] Hong Kong University of Science and Technology (Guangzhou)

## Abstract

In generative modeling, numerous successful approaches leverage a low-dimensional latent space, e.g., Stable Diffusion [68] models the latent space induced by an encoder and generates images through a paired decoder. Although the selection of the latent space is empirically pivotal, determining the optimal choice and the process of identifying it remain unclear. In this study, we aim to shed light on this under-explored topic by rethinking the latent space from the perspective of model complexity. Our investigation starts with the classic generative adversarial networks (GANs). Inspired by the GAN training objective, we propose a novel "distance" between the latent and data distributions, whose minimization coincides with that of the generator complexity. The minimizer of this distance is characterized as the optimal data-dependent latent that most effectively capitalizes on the generator's capacity. Then, we consider parameterizing such a latent distribution by an encoder network and propose a two-stage training strategy called Decoupled Autoencoder (DAE), where the encoder is only updated in the first stage with an auxiliary decoder and then frozen in the second stage while the actual decoder is being trained. DAE can improve the latent distribution and as a result, improve the generative performance. Our theoretical analyses are corroborated by comprehensive experiments on various models such as VQGAN [21] and Diffusion Transformer [60], where our modifications yield significant improvements in sample quality with decreased model complexity.

## 1  Introduction

In the past decade, deep generative models have achieved great success across various domains such as images, audio, videos, and graphs [32, 59, 33, 34]. The advances are epitomized by recent breakthroughs in text-driven image generation [69, 66, 65, 68]. Behind the empirical success are fruitful developments of a wide variety of deep generative models, among which, many can be associated with a latent space, e.g., generative adversarial networks (GANs) [23, 6, 40], Variational Autoencoders (VAEs) [44], and latent diffusion models [79, 68, 60]. It is commonly believed that for structured data such as images, the intrinsic dimension is much lower than the actual dimension [4, 62]. By utilizing a proper low-dimensional latent space, generative modeling can be more efficient [79, 50, 78]. Take text-to-image generation as an example, state-of-the-art models such as DALL-E [65], Parti [88], Stable Diffusion [68] all utilized discrete Autoencoders to alleviate the computation cost by downsampling images, e.g., from 256×256 to 32×32.

In practice, the choice of the latent plays a critical role. Stable Diffusion [68] employs the Autoencoder from VQGAN [21], which is an improvement over VQVAE [80] by incorporating adversarial training.

---

*Corresponding to: Jiacheng Sun (sunjiacheng1@huawei.com)

37th Conference on Neural Information Processing Systems (NeurIPS 2023).

The discrete Autoencoder was later substituted to continuous ones regularized with Kullback–Leibler (KL) divergence. ViT-VQGAN [87] further boosted VQGAN by using vision transformers [20] as the backbone and it was adopted by Parti [88]. Despite the methodological advances, our theoretical understanding of the latent space is still limited. Important questions such as what constitutes a good latent space, what is the optimal choice, and how it depends on data remain elusive.

Ideally, the latent distribution $P_z$ should preserve the important information about the data distribution $P_x$ as much as possible so that the required effort from the model learning is minimal. Finding such a data-dependent latent can be viewed as a self-supervised learning (SSL) task. Can we borrow insights from the vast literature on SSL to understand and improve the latent? Unfortunately, existing methods [57, 14, 25, 28, 29] are designed for discriminative tasks and not suited for generative modeling. To illustrate, consider the classic GANs [24, 76] where the latent distribution is usually pre-defined. When modeling the CIFAR-10 dataset [45], if we substitute the DCGAN's latent [63] from standard Gaussian to the features learned from SimCLR [14] with the same dimension, the Inception Score (IS) drops from 5.68 to 3.93[2]. Therefore, new theoretical insights are in dire need to elucidate the ideal $P_z$ and uncover its connection to $P_x$.

In this work, we aim to provide a new understanding of the latent space in generative modeling from the angles of SSL and minimizing model complexity, where we first formalize the problem for GAN models and then generalize it to Autoencoders.

Drawing inspiration from the training objective of GANs, a novel distance between distributions in different dimensions can be defined to measure the closeness between the latent and the data (Section 3.1). As is typically emphasized in learning theory [82, 83, 35, 61], the complexity of the generator is an important factor in this formulation, where a latent is deemed closer to the data if a generator can achieve the same (better) performance with lower (the same) complexity. The latent closest to the data in a GAN-induced distance is characterized as the optimal choice for that GAN, which leads to the simplest map between the latent and the data.

With our formulation of the optimal latent distribution, the immediate question is how to estimate it, which gives rise to a new SSL task for generative modeling. Naturally, we consider utilizing an encoder network to parameterize the latent, which uncovers the popular Autoencoder architecture with the paired generator as a decoder (Section 3.2). In the learning process, we further emphasize the importance of a relatively weak decoder and the trade-off between the informativeness of the latent and the capability of the decoder (Section 4.1). To this end, we propose a 2-stage training scheme called Decoupled Autoencoder (DAE), where the encoder is only updated in the first stage with an auxiliary decoder and then frozen in the second stage while the actual decoder is being trained (Section 4.2). Our theoretical analyses are corroborated by comprehensive experiments on both synthetic data and real image data with various models such as DCGAN [63], VAEGAN [47, 90], VQGAN [21], and Diffusion Transformer (DiT) [60], where our modifications yield significant improvements in image quality with decreased model complexity (Section 6).

## 2 Preliminary

**Notations.** For a function $f : \Omega \to \mathbb{R}$, let $\|f\|_p = (\int_\Omega |f(\boldsymbol{x})|^p d\boldsymbol{x})^{1/p}$. For a vector $\boldsymbol{x}$, $\|\boldsymbol{x}\|_p$ denotes its $l_p$-norm. $L_p$ and $\ell_p$ are used to distinguish function norms and vector norms. For two positive sequences $\{a_n\}_{n\in\mathbb{N}}$ and $\{b_n\}_{n\in\mathbb{N}}$, we write $a_n \lesssim b_n$ if there exists a constant $C > 0$ such that $a_n \leq Cb_n$ for all sufficiently large $n$. We use $P_x$ and $p_x$ to denote the probability distribution and density of the random variable $x$. $g \circ f(x) := g(f(x))$ denotes composition of functions.

**Generative adversarial networks** are a type of implicit generative models that aim to learn transformations $g \in \mathcal{G}$ from random noises $\boldsymbol{z}$ to the data $\boldsymbol{x}$. At the population level, its objective function can be expressed in general as

$$\inf_{g\in\mathcal{G}} D_h(P_x, P_{g(z)}), \ \boldsymbol{z} \sim P_z, \tag{2.1}$$

where $P_z$ is usually pre-determined to be some simple distribution such as (mixture) Gaussian or uniform, and $D_h$ is realized by the adversarially-trained discriminator $h$ and when optimized properly, can give rise to various well-established distances, e.g., Jensen-Shannon divergence [24],

---

[2]Calculated using reconstructed data. More details can be found in Section 6.2

maximum mean discrepancy [48], $f$-divergence [58], Wasserstein distance [5], etc. Exploiting the flexible architectures of various neural networks as generators [41, 42], GAN enjoys state-of-the-art performance in generating various data types [41, 85, 74, 89] with notable advantage in the sampling speed, especially compared with diffusion probabilistic models (DPMs) [72, 32]. However, GAN models often suffer from mode collapse, failing to keep the data diversity intact. Besides the usual suspect of the min-max adversarial training, the poor choice of latent noise distribution is also to blame [22].

**Self-supervised learning.** The goal of SSL is to learn a (low-dimensional) feature representation $f(\boldsymbol{x})$ from unlabeled data that is suited for various discriminative downstream tasks, e.g., classification, detection, etc. There are mainly three categories, i.e., pretext task-based [57], contrastive methods such as SimCLR [14], MoCo [28], BYOL [25], SimSiam [15], and generative-based such as Masked Autoencoder (MAE) [29]. The learned feature map $f(\boldsymbol{x})$ defines a distribution over the feature space $\boldsymbol{z} \in \mathbb{R}^{d_z}$, denoted as $P_f$. Ideally, $P_f$ should preserve the important information about $P_x$ as much as possible. In this sense, representation learning in general can be seen as preserving certain "distances[3]" between distributions in different dimensions [36, 8].

**Distance between distributions in different dimensions.** To quantitatively measure the closeness, various similarity measures between $P_f$ and $P_x$ are defined. Since $d_z < d$, typical distances between distributions will not work due to the different metric spaces $\boldsymbol{z}$ and $\boldsymbol{x}$ reside in. Fortunately, there are two existing tools we can turn to. The first is Gromov-Wasserstein distance [56, 71], which circumvents the metric spaces mismatch by comparing pairwise joint distributions. Typical manifold learning methods such as stochastic neighbor embedding can be thought of as a special case [31, 81, 36]. Second, as in [9], popular distances such as Wasserstein-$p$ distance and $f$-divergence can all be naturally extended to such settings, either by embedding $P_z$ to $\mathbb{R}^d$ or projecting $P_x$ to $\mathbb{R}^{d_z}$ and taking the infimum over all orthogonal linear projections. Specifically, for any distance between distributions $D(\cdot, \cdot)$ in the Wasserstein-$p$ or $f$-divergence family, define its generalized version to be

$$D^+(P_x, P_z) := \inf_{P_{\widehat{x}} \in \Phi^+(P_z, d)} D(P_x, P_{\widehat{x}}),$$

where $\Phi^+(P_z, d) := \{P_{\widehat{x}} : \boldsymbol{A}\widehat{x} + \boldsymbol{b} \sim P_z, \boldsymbol{A} \in \mathbb{R}^{d_z \times d}, \boldsymbol{b} \in \mathbb{R}^{d_z}, \boldsymbol{A}\boldsymbol{A}^\top = \mathbb{I}_{d_z}\}$ is the family of distributions in $\mathbb{R}^d$ embedded from $P_z$ by a linear orthogonal transformations from $\mathbb{R}^{d_z} \to \mathbb{R}^d$. Similarly, we can define $D^-(P_x, P_z)$ and [9] showed that $D^+(P_x, P_z) = D^-(P_x, P_z)$.

Contrastive learning has deep connections with manifold learning [8] in terms of preserving pairwise similarity, which can be viewed as a special case of preserving the Gromov-Wasserstein distance [36]. In contrast, as we will demonstrate in this work, the optimal latent for generative modeling such as GANs is more closely related to $D^+(P_x, P_z)$.

## 3 Optimal latent distribution for generative models

Recall the typical training objective of GANs (2.1) and denote its empirical version as $L_n(g|P_z)$, where $n$ is the sample size. For any given $P_z$, with limited capacity on the generator, $\min_g L_n(g|P_z)$ reflects the compatibility or closeness between $P_z$ and $P_x$. If $P_{z_1}$ is a better latent than $P_{z_2}$, intuitively $\min_g L_n(g|P_{z_1}) \leq \min_g L_n(g|P_{z_2})$. This can serve as an empirical and relative evaluation of the quality of GAN latents. Inspired by this, we can generalize the distance proposed in [9] to construct a more meaningful version specifically for implicit generative models such as GANs.

### 3.1 GAN-induced distance

Let $\mathcal{M}(\Omega)$ denote the set of all Borel probability measures on $\Omega$. For any distance between distributions $D(\cdot, \cdot)$ in the data space $\mathcal{M}(\mathbb{R}^d)$, define the generalized version associated with a generator $g \in \mathcal{G}$ mapping from $\mathbb{R}^{d_z}$ to $\mathbb{R}^d$ as:

$$D^{\mathcal{G}}(P_z, P_x) := \inf_{g \in \mathcal{G}} D(P_{g(z)}, P_x). \tag{3.1}$$

---

[3] We use the word "distance" quite generally, including not only well-defined metrics, but also divergences.

Compared with $D^+$ in [9], $D^{\mathcal{G}}$ substitutes the linear orthogonal mappings to a general function space. Although $D^{\mathcal{G}}$ is not a valid metric as it is defined between different probability measure spaces, it can serve as a viable measurement of the closeness between $P_z$ and $P_x$.

In order for $D^{\mathcal{G}}$ to be nontrivial, the function space $\mathcal{G}$ cannot be too large, as it is known that one-dimensional distributions can approximate higher dimensional distributions arbitrarily well by neural network transformations in Wasserstein-$p$ distance [86]. On the other hand, $\mathcal{G}$ cannot be too small, or it may not be able to adequately extract the information from the latent distribution. In this work, we consider $\mathcal{G}$ to be a family of neural networks with bounded complexity, i.e., $\mathcal{G}_c = \{g \in \mathcal{G} : C(g) \le c\}$, where $c > 0$ and $C(\cdot)$ is some complexity measure to be specified. $C(\cdot)$ can be as specific as the Lipschitz constant, or as general as the size (width, depth, etc.) of the network.

*Remark* 3.1. It is worth emphasizing that in (3.1), we do not consider the optimization problem but instead, focus on the existence or the global minimum of $\inf_{g \in \mathcal{G}}$. A toy example is given in Appendix A.1 to illustrate the global minimizers of $g \in \mathcal{G}$.

*Remark* 3.2 (Complexity scaling law). In the toy case of Appendix A.1, the relationship between $D^{\mathcal{G}_c}$ and $c$ is an embodiment of the popular *scaling law* phenomenon [39, 91] with respect to the model complexity. It's worth noting that the requirement on the complexity can be drastically decreased if we consider multiple steps of generation, rather than single-step push-forward transformations. From this perspective, one of DPM's main advantages may just be the increased complexity due to the recursive sampling process.

Directly following the definition in (3.1), we have the following propositions.

**Proposition 3.3.** *Let $D$ be a $p$-Wasserstein metric, a Jensen-Shannon divergence, a total variation, or an $f$-divergence. Then, $D^{\mathcal{G}}(P_z, P_x)$ in (3.1) is zero if and only if there exists $g^* \in \mathcal{G}$ such that $P_{g^*(z)} = P_x$.*

**Proposition 3.4.** *For two distributional distances, generalizing them across different dimensions via (3.1) does not change their relative relationships, e.g., if $D$ is Wasserstein distance, we still have $W_p^{\mathcal{G}}(p_x, p_z) \le W_q^{\mathcal{G}}(p_x, p_z)$, for $p \le q$.*

Equipped with $D^{\mathcal{G}}$ to compare different latent distributions, we can characterize the ideal $P_z$ as the one that minimizes $D^{\mathcal{G}}(P_z, P_x)$, i.e., the latent that requires the lowest generator capacity for $L(g|P_z)$ to be below a certain threshold[4]. Such a latent depends on both the data and the GAN training algorithm and can be seen as an SSL task, targeting specifically minimizing the required capacity of the corresponding generator, thus easing the training of the generator.

*Remark* 3.5 (Dimensional collapse). In contrast to the common *dimensional collapse* phenomenon [38] found in contrastive learning where the learned feature distribution is only supported on a low-dimensional subspace, minimizing $D^{\mathcal{G}}(P_z, P_x)$ with respect to $P_z$ encourages it to maintain the intrinsic dimension of the data so that $D(P_g(z), P_x)$ is not ill-posed. The dimensional collapsed latent contributes to the poor performance of SimCLR+DCGAN mentioned in Section 1.

*Remark* 3.6 (Change-of-scale problem). Although $D^{\mathcal{G}}$ serves as a well-defined measurement of closeness between $p_z$ and $p_x$, the latent $p_z$ that minimizes $D^{\mathcal{G}}(p_z, p_x)$ might be ill-posed. If the complexity is not invariant to scaling, e.g., $L_p$ norm, Lipschitz constant, etc., $D^{\mathcal{G}}(P_z(\lambda z), P_x)$ is non-increasing with $\lambda$. In the next section, we impose extra constraints to circumvent this problem, and the optimal $P_z^*$ is characterized therein.

### 3.2 Learning the latent by an encoder

Now that we have characterized the ideal latent distribution given the generator family and the data, the next question is how to find such a good latent. Casting it as an estimation problem, the first decision to make is how to parameterize $P_z$. We could adopt Gaussian mixtures as in [84]. Nonetheless, a more universal and powerful way is to employ an encoder network $f \in \mathcal{F} : \mathbb{R}^d \to \mathbb{R}^{d_z}$, similar to SSL methods. With $P_z$ parameterized by an encoder, the overall structure is an Autoencoder. In the remaining part, we also refer to the generator as the decoder and use the two words interchangeably.

At the population level, we want to match the distributions of $x$ and the reconstructed $g \circ f(x)$, i.e.,

$$\inf_{f \in \mathcal{F}, g \in \mathcal{G}} D(P_x, P_{g \circ f(x)}) = \inf_{f \in \mathcal{F}} \left( \inf_{g \in \mathcal{G}} D(P_x, P_{g \circ f(x)}) \right) = \inf_{f \in \mathcal{F}} D^{\mathcal{G}}(P_x, P_{f(x)}).$$

---

[4]More mathematical ground for this argument can be found in Appendix A.3.

Looking at the encoder and the decoder together, we can see that *optimizing $D(P_{g \circ f}, P_x)$ amounts to learning an encoder $f$ that minimizes $D^{\mathcal{G}}$*. This strategy is employed by VQGAN [21], where the encoder and decoder are optimized together with the discretized latent code. Inspired by this observation, we can characterize the optimal latent distribution $P_z^*$ for a given $P_x$ from the perspective of minimizing the distance between distributions in different dimensions by defining $P_{f*}$ as

$$f^* = \underset{f \in \mathcal{F}}{\arg\min} \, D^{\mathcal{G}}(P_x, P_{f(x)}). \tag{3.2}$$

Notice that $P_f^*$ not only depends on $P_x$ and $D$, but also $\mathcal{F}$ and $\mathcal{G}$. The change-of-scale problem in Remark 3.6 is mostly taken care of by the inherent boundedness of $\mathcal{F}$. More discussion on the choice of $\mathcal{G}$ can be found in Section 4.1.

*Remark* 3.7 (Classification). As the solution of a new SSL task, $f^*$ in (3.2) will preserve well-separated clusters of $P_x$, though not as discriminative as the features learned from existing SSL methods. See Proposition A.3 in the appendix for more details. We also conducted experiments on simulated data evaluating the classification accuracy in Section 6.1.

With the encoder, (3.2) is not the only option to characterize the optimal latent distribution. Besides directly parameterizing $P_z$, $\mathcal{F}$ can also serve an auxiliary role in defining a more general $P_z^*$. To this end, in parallel to $D^{\mathcal{G}}$, we can also define $D^{\mathcal{F}}(P_z, P_x) := \inf_{f \in \mathcal{F}} D(P_z, P_{f(x)})$. As $D^{\mathcal{G}}$ is to $D^+$, $D^{\mathcal{F}}$ generalizes the $D^-$ in [9]. Combining $D^{\mathcal{G}}$ and $D^{\mathcal{F}}$, we get a generalized measure of distance between the latent and data for Autoencoders as $D^{\text{æ}}(P_z, P_x) = D^{\mathcal{G}}(P_x, P_z) + D^{\mathcal{F}}(P_x, P_z)$, which resembles the *condition number* of matrices, but for latent distributions. The optimal latent distribution can also be characterized by

$$P_z^* = \underset{P_z}{\arg\min} \, D^{\text{æ}}(P_z, P_x). \tag{3.3}$$

If $d_z = d$, then identity mapping, i.e., $P_z^* = P_x$, obviously solves (3.2) and (3.3). If like DPMs or VAEs where the latent distributions are chosen to be data-agnostic, e.g., standard Gaussian, $D^{\text{æ}}(P_z, P_x)$ can be significantly larger for complicated data.

*Remark* 3.8. $D^{\mathcal{G}}$ and $D^{\mathcal{F}}$ can be zero, especially when $P_x$ is supported on a low-dimensional space [9]. As a result, the defined $f^*$ or $P_z^*$ might not be unique. We provide a toy example in Appendix A.2 to illustrate how $P_z^*$ minimizes the complexity.

# 4 Improving the latent with Decoupled AutoEncoder

In practice, the encoder and decoder family should be as powerful as possible. However, as stated before, $\mathcal{G}$ and $\mathcal{F}$ should be chosen carefully in order for the induced $D^{\mathcal{G}}$ and $D^{\mathcal{F}}$ to be meaningful.

## 4.1 Necessity of relatively weak decoders

Given a total budget[5] of $\mathcal{G}$ and $\mathcal{F}$, intuitively, $C(\mathcal{G}) = C(\mathcal{F})$ tend to result in the best approximation of $\mathcal{G} \circ \mathcal{F}$ to the identity mapping. This is reflected in practice where the encoder-decoder pair is often designed with symmetric architecture, with the same number of blocks and parameters, e.g., up-sampling vs. down-sampling, convolution vs. deconvolution, etc. However, when it comes to the quality of the encoded latent, only targeting the reconstruction error may not be the ideal strategy. As we will demonstrate in Section 4.2, a better way is to first target a good latent distribution, and then do reconstruction using the learned latent. Consider two symmetric cases: (larger encoder, smaller decoder) vs (smaller encoder, larger decoder). The reconstruction ability may not be significantly different, although both are sub-optimal, the former tends to produce better latent distribution. See Section 6.1 for empirical evidence in the toy case comparing the two cases.

For a concrete demonstration, consider the simple linear case where the encoder and the decoder are two matrices, denoted as $W_e \in \mathbb{R}^{d \times d_z}$ and $W_d \in \mathbb{R}^{d_z \times d}$ respectively. Take the matrix 2-norm as a measurement of complexity, which is equal to the largest singular value. A similar setting is also considered in [37]. In the linear case, the quality of the latent is equivalent to the reconstruction error and the optimal encoder should recover the principal components. The next theorem states that the Autoencoder could fail if the encoder is not large enough.

---

[5] The budget and $C(\cdot)$ here refer to the general concept of size (width, depth, etc.) of the network.

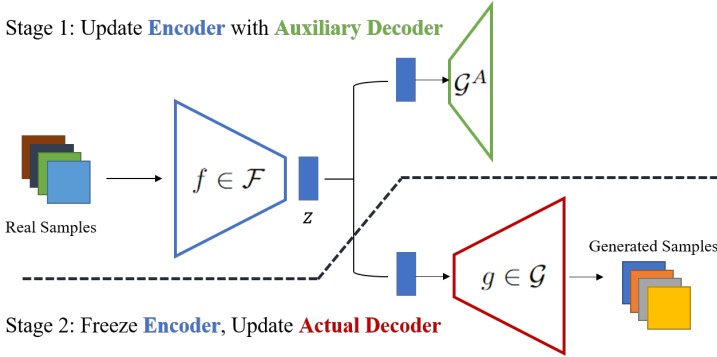

Stage 1: Update **Encoder** with **Auxiliary Decoder**

Real Samples

$f \in \mathcal{F}$

$z$

$\mathcal{G}^A$

$g \in \mathcal{G}$

Generated Samples

Stage 2: Freeze **Encoder**, Update **Actual Decoder**

Figure 1: Overview of the DAE 2-stage training. DAE addresses the trade-off between generation quality and the latent quality where the encoder is only updated in the first stage with an auxiliary weaker decoder and then frozen in the second stage while the actual decoder is being trained.

**Theorem 4.1.** *Let the singular values of $W_d$ be $\lambda_1 \geq \ldots \geq \lambda_{d_z}$. If $\|W_e\| < 1/\lambda_{d_z}$, the linear Autoencoder cannot recover the principal components, and the reconstruction error is sub-optimal.*

The linear case provides insights into neural networks as Autoencoders, indicating that the encoder should be *relatively strong* to better learn the latent distribution. In the more practical nonlinear cases, things become more complicated. The intuition that (larger encoder, smaller decoder) is preferred to (smaller encoder, larger decoder) for learning a more informative latent is supported by the well-known *posterior collapse* phenomenon in the VAE literature [1, 27, 67, 54], where the encoded distribution becomes completely uninformative if the decoder is too powerful. However, for implicit generative models such as GANs, the generator is obviously the key component and it should be as powerful as possible. There exists a *trade-off* between generation quality and the quality of the latent. Using a decoder too powerful can hinder the learning process of a good latent while a decoder too small will hurt the generation performance. In practice, whether a decoder family is weak or not can be summarized in an oversimplified way by its size, i.e., a smaller decoder is weaker.
*Remark* 4.2. We could consider pairing a sufficiently large $\mathcal{G}$ with an even larger $\mathcal{F}$. Unfortunately, such a combination may not be a good solution. Besides the added computational cost, an encoder-decoder pair too powerful can drive $D^{\mathcal{G}} + D^{\mathcal{F}}$ too close to zero, so that it may get overwhelmed by random noise. The signal-to-noise ratio may be too low for effective optimization.

## 4.2 Decoupled AutoEncoder

To address the trade-off between generation quality and the quality of the latent, we can introduce an auxiliary decoder $\mathcal{G}^A$, potentially much smaller/weaker, to substitute $\mathcal{G}$ for training the encoder. Once the encoder is trained, we can then freeze it and pair it with $\mathcal{G}$ for the decoder training. The key idea is to decouple the learning process of the encoder and the decoder in a 2-stage training scheme we call **Decoupled AutoEncoder** (**DAE**). Figure 1 illustrates the method overview where the two stages of DAE can be summarized as:

- DAE Stage 1: Train $\widehat{f} \in \mathcal{F}$ with a small decoder $\mathcal{G}^A$ to learn a good latent.
- DAE Stage 2: Freeze the trained encoder $\widehat{f}$. Then train the regular decoder $g \in \mathcal{G}$ to ensure good generation performance.

By the 2-stage training, one with $\mathcal{G}^A$ and one with $\mathcal{G}$, without much hyperparameter tuning, DAE can learn a better latent distribution and as a result, achieve better reconstruction and generative modeling.
*Remark* 4.3 (Realizing $\mathcal{G}^A$). The essence of the first stage is to employ an auxiliary decoder that is relatively weak compared with the encoder. Straightforwardly, we can introduce an extra decoder family. For an easier way of implementation, we can also use various forms of regularization to realize $\mathcal{G}^A$. For instance, with a balanced encoder-decoder pair, we can apply strong Dropout [73] to the decoder in the first stage to make it weaker. Both methods are experimented in Section 6.

DAE can be applied in a **plug-and-play** style to any existing Autoencoder training pipelines with the decoupled 2-stage modification. We demonstrate in Section 6 that compared to various baselines,

e.g., VAE, VQGAN, etc., our DAE-modified versions are better in terms of both reconstruction error and quality of the latent. For instance, we refer to Figure 3 in the appendix where we experimented with VQGAN [21]. In terms of the reconstruction error, Stage 1 is worse than the baseline, which is to be expected since a weaker decoder is used. However, paired with the Stage 1 encoder, the decoder (the same size as the baseline) can achieve better reconstruction performance.

### 4.3 Sampling from the latent distribution

In most Autoencoder-based generative models, e.g., VAE, adversarial Autoencoder [55], Wasserstein Autoencoder [76], etc., the latent distributions are all modeled by Gaussian distributions. DPMs [72, 32] can also be seen as a type of Autoencoder-based method with the forward noising process as the encoder and the denoising process as the decoder, with the latent distribution being Gaussian. As discussed in Remark 3.2, the decoding process of DPM enjoys drastically increased capacity due to the multi-step generative process.

In our work, with $P_z$ parameterized by the encoder, how to generate new samples from it requires extra modeling of the latent. Otherwise, we can only do reconstruction. There are two mainstream methods to sample from the latent distribution. The first is inspired by natural language processing. In VQGAN [21] and Parti [88], the discretized latent codes are autoregressively modeled by transformers as a sequence, similar to GPT [64] while masked image modeling methods [11, 12] resembles BERT [19]. The second is by diffusion process. Latent space diffusion models [68, 60] can be seen as using DPMs to model the latent space of a powerful Autoencoder. We mainly adopt the VQGAN formulation in our numerical experiments for training the Autoencoders. To validate the effectiveness of our DAE, we consider modeling the latent by both transformers and diffusion models.

## 5 Related works

BourGAN [84] proposed to model the latent distribution as Gaussian mixtures as a remedy for the mode collapse problem of GANs. Instance-conditioned GAN [10] proposed to partition the data manifold into a mixture of overlapping neighborhoods described by a data point and its nearest neighbors to improve the quality of unconditional generation. There is also a line of work that considers post-sampling latent space mining of GANs by exploiting the trained discriminator [75, 13]. [49] further proposed adding an implicit latent transform before the mapping function to improve the latent from its initial distribution in a bi-level optimization fashion. Though these works targeted improving the latent distribution of GANs, they did not characterize the optimal choice nor employ an encoder to model the latent. In comparison, we approach the problem from the perspective of preserving the distance between distributions in different dimensions and propose a 2-stage training scheme involving an encoder.

There are GANs that incorporate an Autoencoder structure. VAEGAN [47, 90] combines a VAE by collapsing the decoder and the generator into one. [2] utilized a standard autoencoder to embed data into the latent space to address the mismatch of the continuous neural network and the discontinuous optimal transport (OT) map. [3] later proposed to utilize semi-discrete OT maps to sample from the latent and train GAN models. VQGAN [87] uses vector-quantized latent codes and transformers to model the tokenized latent distribution. ViT-VQGAN [87] later made improvements over VQGAN. Besides using vision transformers [20] as the backbone, it also introduced an $l_2$ regularization term to the latent code. However, the aforementioned works did not characterize the optimal latent either and did not point out the necessity of the decoupled 2-stage training. Our DAE approach can also be applied to them for further improvement.

In the literature of VAE, [77] introduced new priors consisting of mixture distribution with components given by variational posteriors conditioned on learnable pseudo-inputs. [17] proposed a two-stage learning scheme to amend the problems associated with the Gaussian encoder/decoder assumptions. [1] stated that obtaining a good ELBO is not enough for good representation learning and proposed to incorporate mutual information and rate-distortion curve to achieve a better balance between the informativeness of the latent and the reconstruction quality. In comparison, our characterization of the optimal latent is from the perspective of matching distributions and minimizing complexities and is not restricted to Gaussian assumptions. It is also worth emphasizing that while the mutual information, closely related to VAEs, can also serve as a "distance" between distributions in different dimensions, it will not work for our benefit since we are considering *deterministic* encoders and

decoders. In the deterministic case, the conditional entropy would be zero and the mutual information would always be the entropy of the latent. Hence uniformly distributed latent is always ideal, which does not offer new insights.

Masked Autoencoder (MAE) [29] is a generative-based SSL method with the training objective of reconstructing masked images. In MAE, a high mask ratio and a relatively weak decoder have been verified as important factors for good discriminative performance, which is also corroborated by our perspective of preserving distributions in different dimensions. MAEs are designed for discriminative tasks and their generation capability is yet to be explored.

## 6 Experiments

In this section, we conduct empirical evaluations of our proposed DAE training scheme on a variety of datasets and generative models, from toy Gaussian mixture data to DCGAN on CIFAR-10, to VQGAN and DiT on larger datasets. The detailed experiment settings can be found in Appendix B.

### 6.1 Toy examples

To showcase the effectiveness of the proposed 2-stage training, we consider a toy Gaussian mixture case with 8 clusters, whose centers are evenly placed on the unit circle, and the variances are 0.25. All samples in $\mathbb{R}^2$ are then projected to $\mathbb{R}^{10}$ via a linear orthogonal mapping. We employ the vanilla VAE [44] for demonstration with $d_z = 2$. In our implementation, both the encoder and the decoder are 3-layer fully connected networks with different hidden dimensions to control the size. Giving each cluster a label, we evaluate (1) the nearest neighbor classification accuracy from the 2D latent; (2) the nearest neighbor classification accuracy from the 10D reconstruction; and compare different configurations of the encoder and decoder pair. The average classification accuracy is reported in Table 1, which corroborates our analysis in Section 4, i.e., a relatively weak decoder can learn a better latent and the following 2-stage training can further improve reconstruction.

Table 1: The mean (variance) of the classification accuracy w.r.t. difference hidden dimensions of the (encoder, decoder) over 10 replications.

| Hidden Dim | (64, 128) | (128, 64) |
|---|---|---|
| Latent Acc (%) | 80.6 (7.0) | **87.8** (5.8) |
| Hidden Dim | (128, 128) | DAE (128, 128) |
| Rec Acc (%) | 92.2 (6.1) | **98.0** (1.3) |

Table 2: The performance of DCGAN with different latent distributions.

| Method | IS ($\uparrow$) | FID ($\downarrow$) |
|---|---|---|
| DCGAN (reproduced) | 5.68 | 51.76 |
| DCGAN-SimCLR | 3.93 | 168.23 |
| VAEGAN | 5.82 | 48.11 |
| **DAE-VAEGAN** | **6.16** | **46.12** |

### 6.2 GAN on CIFAR-10

As a more comprehensive proof-of-concept, we experiment with DCGAN [63] on the CIFAR-10 [45] dataset. Specifically, we consider a vanilla baseline where the generator has 5 convolutional layers and the discriminator has 4 convolutional layers, followed by a linear classification layer. The latent for DCGAN is standard Gaussian. To corroborate our analysis in Section 4, we adapted the DCGAN with varying latent space configurations: (1) We utilize a latent space defined by a CIFAR-10 pre-trained SimCLR model [16], denoting this adaptation as DCGAN-SimCLR. (2) We consider VAEGAN with a learnable latent space. The structures of the decoder and discriminator mirrored those of our baseline DCGAN while the encoder is implemented with a five-layer CNN. (3) We modified VAEGAN to DAE-VAEGAN, where the width of convolutional layers in the decoder is halved in the first stage and then set back to its original structure in the second stage.

We evaluated the aforementioned four methods using the metrics of Inception Score (IS) [70] and Frechlet Inception Distance [30]. With the exception of DCGAN, which generates images from random noises, the other methods first encode an image into latent space, and then generate an image based on that encoding. To this end, we omit the latent space modeling and calculate their metrics from reconstructed images. The results are listed in Table 2, with two take-home messages: (1) Traditional SSL methods may not help with the latent for generative modeling; (2) Decoupling the encoder and decoder training can improve the overall performance of generative modeling.

## 6.3 VQGAN

VQGAN [21] consists of two major components. The first is a discrete Autoencoder, where the latent is quantized by looking up nearest neighbors from a learnable codebook. After the encoder and decoder are trained (objective contains adversarial loss), the second component, a transformer model, learns to generate discrete codes autoregressively in the latent space, which are then mapped to images by the learned decoder.

To demonstrate the effectiveness of our DAE, we introduce the decoupled training strategy to the first component of the standard VQGAN. In the first stage, we implement the relatively weak auxiliary decoder by applying Dropout [73] to the decoder with ratio $p$, and train encoder and the dropped decoder jointly[6]. Then in the second stage, we freeze the encoder and train the full decoder without Dropout. Training of the transformer component is the same as VQGAN. The modified model is denoted as DAE-VQGAN.

**Setup.** We evaluate DAE-VQGAN by comparing with the baseline VQGAN on the FacesHQ dataset, which is a combination of two face datasets CelebAHQ [51] and FFHQ [43], with 85k training images and 15k validation images in total. In all experiments, the input image size is 256x256. We use the official VQGAN implementation[7] and model architectures for FacesHQ. The Autoencoder is dubbed VQ-f16. The training process of DAE-VQGAN is divided equally into two stages, where in the first stage we apply 2D Dropout (channel-wise) to the decoder with ratio $p = 0.5$.

Table 3: Reconstruction FID over FacesHQ training and validation sets, and transformer generation FID on CelebAHQ and FFHQ training sets. †: Evaluated on the publicly available pre-trained model on FacesHQ. *: Our reproduction is based on the official VQGAN implementation.

| Method | Reconstruction | | Generation | |
|---|---|---|---|---|
| | Train | Val | CelebaHQ | FFHQ |
| VQGAN | 4.81† | 6.27† | 10.2 | 9.6 |
| VQGAN* | 4.23 | 5.83 | 9.97 | 10.44 |
| DAE-VQGAN | **2.01** | **3.82** | **8.58** | **8.36** |

Table 4: Class-conditional image generation on ImageNet $256 \times 256$ using the DiT-L/4 model with different Autoencoders. *: Our reproduction is based on the official DiT implementation, which is consistent with the results reported in Fig.5 of [60].

| Method | FID | sFID | IS |
|---|---|---|---|
| DiT* | 35.16 | 7.33 | 39.25 |
| **DAE-DiT** | **32.29** | **6.90** | **41.71** |

**Improved image reconstruction and generation.** To quantitatively evaluate the encoder-decoder component, we train the model on FacesHQ and compute the reconstruction FID over the full training and validation splits. We further train a transformer separately for CelebAHQ and FFHQ with the same architecture as in [21], and compute the generation FID with 50k generated samples against the training split of the respective dataset. As shown in Table 3, DAE-VQGAN achieves significantly improved FID compared to VQGAN, indicating that a stronger encoder-decoder component can be unlocked for VQGAN by our DAE approach. We show qualitative generation results in Figure 2. We see that models trained by DAE-VQGAN can reconstruct and generate images with high fidelity. Figure 3 in the Appendix shows the reconstruction losses during training. It is worth noting that while our DAE-modified VQGAN has higher reconstruction losses in the first stage, it catches up fast in the second stage and converges to a lower level than the baseline. This is to be expected as the first stage of DAE focuses on learning a better latent, which can ease the training of the decoder in the second stage and result in better performance overall.

**Decreased Complexity.** Minimizing the model complexity is a key motivation for our DAE. To verify our claims, we evaluate the Lipschitz complexity ($C_{Lip}$ defined in (B.1)) of the encoder $f$ and decoder $g$ in VQGAN. Since $g \circ f$ is trained to be approximately the identity map, the trends of $C_{Lip}(f)$ and $C_{Lip}(g)$ are expected to be the opposite, with the ideal number being close to 1 for both if we want the overall complexity $C_{Lip}(f) + C_{Lip}(g)$ to be minimized. We observed in Figure 4 that the encoder complexity of VQGAN and DAE-VQGAN are 1.40 and 0.88, respectively, with an increasing trend in terms of training steps; the decoder complexity of VQGAN and DAE-VQGAN

---

[6]We also implemented the auxiliary weak decoder by leveraging an extra decoder family with the number of convolution channels halved. The overall performance is similar and we put the results in Appendix B.1

[7]https://github.com/CompVis/taming-transformers

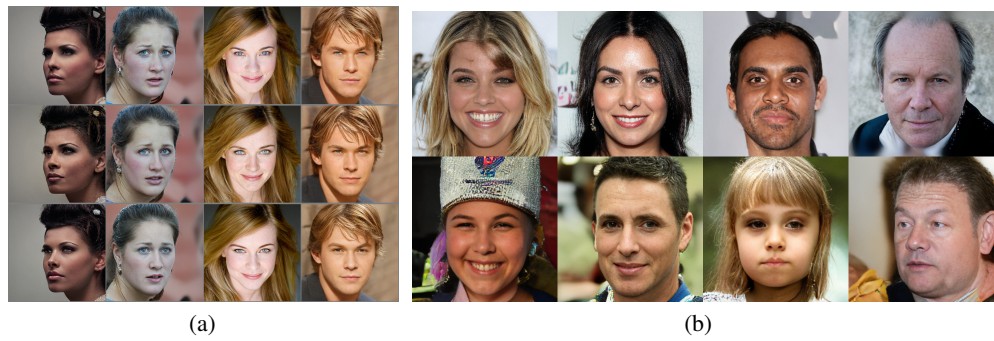

(a)                             (b)

Figure 2: (a) Top: original image; Middle: reconstruction of VQGAN; Bottom: reconstruction of DAE-VQGAN. (b) Generated samples of DAE-VQGAN.

are 0.73 and 1.16, respectively, with a decreasing trend in terms of training steps. Compared with the baseline, our DAE-modified version produces better latent distribution with a significantly lower complexity while achieving a lower reconstruction error. See Appendix B.2 for more details.

### 6.4 Diffusion Transformers (DiT)

The encoder in VQGAN is discrete with a codebook. In Appendix B.3, we look into the codebook and investigate how different training strategies affect the behavior of the learned latent. Our DAE can also apply to continuous Autoencoders that are usually regularized by KL divergence. Such KL-regularized AEs are adopted by latent diffusion models such as DiT [60]. Following similar settings as VQGAN, we also modified DiT to DAE-DiT by employing the decoupled training scheme and experimented with the larger ImageNet dataset [18]. The details can be found in Appendix B.4.

For evaluation, we train DiT on the ImageNet dataset with $256{\times}256$ resolution following the official implementation[8]. Instead of the largest DiT-XL/2 model (675M Params), we select the DiT-L/4 model (458M Params) for higher training efficiency. The corresponding Autoencoder is dubbed KL-f8. AdamW [53] optimizer is employed with a constant learning rate of $10^{-4}$ and a weight decay of $3 \times 10^{-2}$. The batch size is 1024, and the number of epochs is 120. The trained model with Exponentially Moving Average (EMA) is then used to generate 50k images of the 1000 categories equally via a 250-step DDPM sampling [32] and a classifier-free guidance scale of 4. Table 4 compares the image generation quality where we can see that our DAE-DiT achieves significant improvement, similar to what we observed for DAE-VQGAN.

## 7   Discussion

This work investigates the ideal latent distribution for generative models. We introduce a novel distance between distributions to characterize the ideal $P_z$ that minimizes the required model complexity. Practically, we propose a two-stage training scheme called DAE that achieves practical improvements on various models such as GAN, VQGAN, and DiT. Since many of the most powerful generative models are associated with a latent space, the impact of such investigations is potentially very high.

Nevertheless, there are many limitations of our work that call for further research along this line. First, our formulation of the optimal latent distribution is based on $D^{\mathcal{G}}, D^{\mathcal{F}}$ and $D^{æ}$, which serve mainly illustrative purposes. The proposed distances cannot be effectively calculated. Second, the effectiveness of DAE is not proven mathematically and the resulting latent is not guaranteed to be closer to $P_z^*$. Our work could be further strengthened if the aforementioned limitations can be addressed. It would also be an interesting direction to explore how our method affects latent space disentanglement [52, 26].

---

[8]https://github.com/facebookresearch/DiT.

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

# A Technical details

## A.1 Toy example 1

To illustrate the interplay between $D^{\mathcal{G}}$ and $\mathcal{G}$, we devise a one-dimensional toy example where $x \sim N(0, \sigma^2)$, $z \sim U(0,1)$, $g : \mathbb{R} \to \mathbb{R}$ is monotonically increasing and $C(g)$ is the Lipschitz constant upper bounded by $c$. In this case, the optimal transformation is $g^* = P_x^{-1}$, whose derivative around 0 or 1 is exponentially dependent on $\mu/\sigma$. Due to the Lipschitz constraint, by the change of distribution formula, we have

$$p_{g(z)}(x) = \mathbb{I}_{\{g^{-1}(x) \in (0,1)\}} \cdot \nabla g^{-1} \geq \frac{1}{c}.$$

From the perspective of regression, the optimal $\ell_2$-projection of $g^*$ to $\mathcal{G}_c$ is

$$\widehat{g}(z) = \begin{cases} g^*(z) & \text{if } \nabla g^*(z) \leq c, \\ c(z - \frac{1}{2}) & \text{else.} \end{cases}$$

Let $z_c = \sqrt{2\sigma^2 \log(c/2\pi)}$ and it is clear that $\nabla g^*(z_c) = c$, or equivalently, $p_x(z_c) = 1/c$. As a result, the push-forward distribution is a truncation of $P_x$ supported on $|x| \leq z_c + c \cdot P_x(-z_c)$, i.e., $P_{\widehat{g}} = P_x$ if $p_x \geq 1/c$ and $1/c$ otherwise, which is equivalent to minimizing the total variation (TV) between $P_g$ and $P_x$. In this oversimplified example, it is easy to see the scaling between $c$ and $D^{\mathcal{G}_c}$, i.e., $TV^{\mathcal{G}_c}(P_z, P_x) \lesssim P_x(-z_c)$, which is polynomial with $c$ in the Gaussian case.

## A.2 Toy example 2

To illustrate the characterization of $P_z^*$ and how it minimizes the complexity, consider a toy example where $d_z = 1$, $x \sim N(0, \Sigma_d)$ and both $\mathcal{G}$ and $\mathcal{F}$ are linear functions with bounded Lipschitz constant $c > 1$. Denote the singular values of $\Sigma_d$ as $\lambda_1 \geq \ldots \geq \lambda_d$. In this case, we have $C(P_z^*) = 0$ and $P_z^* = N(0, \sigma^2)$ with $\sigma \in (\sqrt{\lambda_n}, \sqrt{\lambda_1})$.

**Lemma A.1** (Example 6.1 in [9])**.** *If $D$ is Wasserstein-2 metric,*

$$D^-(N(0,\sigma^2), N(0,\Sigma)) = \begin{cases} \sqrt{\lambda_n} - \sigma & \text{if } \sigma < \sqrt{\lambda_n}, \\ 0 & \text{if } \sqrt{\lambda_n} \leq \sigma \leq \sqrt{\lambda_1}, \\ \sigma - \sqrt{\lambda_1} & \text{if } \sigma > \sqrt{\lambda_1}, \end{cases}$$

*where $\lambda_1 \geq \cdots \geq \lambda_d$ are the singular values of $\Sigma$.*

As a direct corollary of the above lemma, we have $C(P_z^*) = 0$ and $P_z^* = N(0, \sigma)$ with $\sigma \in (\sqrt{\lambda_n}, \sqrt{\lambda_1})$.

## A.3 Discussion on $P_z^*$ and $C(g)$

The ideal $P_z^*$ is characterized as the one that minimizes $D^G(P_z, P_x)$. The link between its definition and the capacity of the generator can be made mathematically sound with an extra assumption.

**Assumption A.2.** For any $P_z$ and $P_x$, $D^{G_c}(P_z, P_x)$ is continuous and monotonically decreasing with c.

Assumption A.2 is not provable for general $D(\cdot, \cdot)$ and $C(\cdot)$. Nevertheless, we give proof that the ideal $P_z^*$ that minimizes $D^G(P_z, P_x)$ will give rise to the minimal complexity generator if it holds.

*Proof.* First, recall that $G_c := \{g \in G : C(g) \leq c\}$ where $C(g)$ is defined as some complexity measurement. It is easy to see that for any $c_2 \geq c_1 > 0$, $G_{c_1} \subset G_{c_2}$. Therefore, $D^{G_{c_2}}(P_z, P_x) \leq D^{G_{c_1}}(P_z, P_x)$.

Consider a generator family $G$ with bounded complexity and we target to have the generated distribution close to the data at level $\epsilon$ measured by $D$, i.e., $D(P_x, P_{g(z)}) \leq \epsilon$ for some $P_z$ and $g \in G$.

For a certain (non-degenerate) $P_z$, there exists $c > 0$ such that $D^{G_c}(P_z, P_x) = \epsilon$, that is, by using this latent $P_z$, we need at least complexity $c$ to achieve the $\epsilon$ goal. If $P_z$ is not ideal, by definition we know that there exists $P_z^*$ that $D^{G_c}(P_z^*, P_x) = \epsilon^* < \epsilon$. By Assumption A.2, we know there exists $c' < c$ such that $D^{G_{c'}}(P_z^*, P_x) = \epsilon$. This means that the goal can be achieved using a lower complexity generator if the latent is ideal. $\qquad\square$

## A.4 Proof of claims in Section 3

Proposition 3.3 and 3.4 directly follow the definition of $D^{\mathcal{G}}$.

$D^{\mathcal{G}}$ is to $D^{\mathcal{F}}$ as $D^+$ is to $D^-$. Due to the enlarged function space, the equality $D^+ = D^-$ no longer holds. Directly following Lemma 2.1 in [9], we have the following inequality: $D^{\mathcal{G}}(P_z, P_x) \leq D^{\mathcal{F}}(P_z, P_x)$ where $D$ can be a $p$-Wasserstein metric or an $f$-divergence.

**Proposition A.3** (Classification guarantee for clustered data)**.** *Suppose $P_x$ has $m$ disjoint supports. Then the corresponding $P_z^*$ must preserve all the clusters.*

*Proof.* Let $D_{KL}$ be the KL divergence (for example). Suppose $P_x$ has $m$ disjoint supports. Then

$$D_{KL}(P_x, P_{\widehat{x}}) = \sum_{k=1}^{m} \int_{X_k} p_x(x) \log\left(\frac{p_x(x)}{p_{\widehat{x}}(x)}\right) dx.$$

To minimize $D_{KL}(P_x, P_{\widehat{x}})$, clearly $p_{\widehat{x}}(x) = 0$ implies $p_x(x) = 0$. As $p_{\widehat{x}}(x)$ increases, $D_{KL}(P_x, P_{\widehat{x}})$ decreases. Since $P_{\widehat{z}}$ can be selected according to our will, if $p_x(x) = 0$ but $p_{\widehat{x}}(x) \neq 0$, it is possible to build $\widetilde{p}_{\widehat{z}}$ such that $\widetilde{p}_{\widehat{x}}(x) \geq p_{\widehat{x}}(x)$ for all $x$, and $1 = \sum_{k=1}^{m} \int_{X_k} \widetilde{p}_{\widehat{x}}(x) dx > \sum_{k=1}^{m} \int_{X_k} p_{\widehat{x}}(x) dx$, which implies there exists a positive measure such that $\widetilde{p}_{\widehat{x}}(x) > p_{\widehat{x}}(x)$. This implies $p_{\widehat{x}}(x)$ cannot minimize $D_{KL}(P_x, P_{\widehat{x}})$. Thus, we must have if $p_x(x) = 0$, then $p_{\widehat{x}}(x) = 0$. This further implies that $z$ must have at least $m$ disjoint supports. If $z$ has less than $m$ disjoint supports, then $\widehat{x}$ must have less than $m$ disjoint supports, because the linear map is continuous. Then there must exists $p_x(x) = 0$ but $p_{\widehat{x}}(x) \neq 0$.

In [9], it has been proved that $\inf D_{KL}(P_x, P_{\widehat{x}}) = \inf D_{KL}(P_z, P_{\widehat{z}})$. Following the same logic, the number of supports of $\widehat{z}$ must be smaller than the number of supports of $x$, which implies the number of supports of $\widehat{z}$ must be smaller than $m$. Combining these results with the previous one, we must have the number of supports of $\widehat{z}$ is $m$.

$\square$

## A.5 Proof of claims in Section 4

Before we prove Theorem 4.1, we introduce the following lemma on linear auto-encoder.

**Lemma A.4.** *Consider two optimization problems:*

$$\min_{W \in \mathbb{R}^{m \times n}} \|X - WW^T X\|_F^2, \tag{A.1}$$

*which corresponds to Principle Component Analysis (PCA), and*

$$\min_{W_1, W_2 \in \mathbb{R}^{m \times n}} \|X - W_2 W_1 X\|_F^2, \tag{A.2}$$

*which is a linear auto-encoder. Let $W^*$ be the solution to PCA and $W_1^*, W_2^*$ be the solution to linear auto-encoder, and*

$$L_1 = \|X - W^*(W^*)^T X\|_F^2, L_2 = \|X - W_2^* W_1^* X\|_F^2. \tag{A.3}$$

*Then, $L_1 = L_2$.*

*Proof.* Clearly, $L_2 \leq L_1$, because we can set $W_1^T = W_2 = W^*$, which leads to

$$L_2 = \|X - W_2^* W_1^* X\|_F^2 \leq \|X - W^*(W^*)^T X\|_F^2 = L_1.$$

Therefore, it suffices to show $L_1 \leq L_2$.

For a fixed $W_2$, By [7], the optimal solution with respect to $W_1$ is

$$W_1 = (W_2^T W_2)^{-1} W_2^T.$$

Therefore,

$$L_2 = \|X - W_2^*((W_2^*)^T W_2^*)^{-1}(W_2^*)^T X\|_F^2.$$

Let the singular value decomposition of $W_2^*$ be

$$W_2^* = U\Sigma V,$$

where $U \in \mathbb{R}^{n \times n}$ and $V \in \mathbb{R}^{m \times m}$ are orthogonal matrices, and $\Sigma \in \mathbb{R}^{n \times m}$ is a diagonal matrix with singular values $\lambda_1 \geq \ldots \geq \lambda_m$. Therefore,

$$
\begin{aligned}
W_2^*((W_2^*)^T W_2^*)^{-1}(W_2^*)^T X &= U\Sigma V(V^T\Sigma^T U^T U\Sigma V)^{-1}V^T\Sigma^T U^T X \\
&= U\Sigma(\Sigma^T\Sigma)^{-1}\Sigma^T U^T X.
\end{aligned}
$$

Note that $\Sigma^T\Sigma$ is a diagonal matrix. Therefore, we can set

$$\widetilde{W} = U\Sigma(\Sigma^T\Sigma)^{-1/2}.$$

Clearly,

$$(\widetilde{W})^T\widetilde{W} = (\Sigma^T\Sigma)^{-1/2}\Sigma^T U^T U\Sigma(\Sigma^T\Sigma)^{-1/2} = I_m,$$

which implies

$$
\begin{aligned}
L_2 &= \|X - W_2^*((W_2^*)^T W_2^*)^{-1}(W_2^*)^T X\|_F^2 \\
&= \|X - U\Sigma(\Sigma^T\Sigma)^{-1}\Sigma^T U^T X\|_F^2 \\
&= \|X - U\Sigma_1 U^T X\|_F^2 \\
&= \|X - U_m U_m^T X\|_F^2 \\
&\geq \|X - U_m^*(U_m^*)^T X\|_F^2 \\
&= L_1,
\end{aligned}
$$

where $\Sigma_1 = \mathrm{diag}(1, ..., 1, 0, ..., 0)$ is a rank $m$ diagonal matrix. This finishes the proof. $\qquad\square$

With the above lemma, Theorem 4.1 is a simple corollary.

*Proof of Theorem 4.1.* Recall the proof of Lemma A.4 where the linear encoder is $W_1$ and the linear decoder is $W_2$. Then, in order to realize the PCA solution, we have

$$
\begin{aligned}
\|W_1\|_2 &= \|(W_2^T W_2)^{-1}W_2^T\|_2 = \|(\Sigma^T\Sigma)^{-1}\Sigma^T\|_2 = \lambda_m^{-1}, \\
\|W_2\|_2 &= \|\Sigma\|_2 = \lambda_1.
\end{aligned}
\tag{A.4}
$$

Thus, as long as $\lambda_1\lambda_m < 1$, the encoder is larger in terms of 2-norm. $\qquad\square$

# B  More on Experiments

## B.1  Experiment details for DAE-VQGAN

Let's recap the VQGAN basics with more details. VQGAN [21] consists of two major components. In the first component, an input image $x$ is encoded by an encoder $f$ into latent representation $z = f(x)$. A quantization step is applied to obtain $z_q = \tau(z)$, which contains nearest neighbors of entries of $z$ from a learnable codebook. Then a decoder $g$ reconstructs $\hat{x} = g(z_q)$ from the codes. After the encoder and decoder are trained (with adversarial loss), the second component, a transformer model, learns to generate discrete codes in the latent space, which are then mapped to images by the learned decoder $g$.

We evaluate our DAE modifications to VQGAN on the FacesHQ dataset, which is a combination of two face datasets CelebAHQ and FFHQ, with 85k training images and 15k validation images in total (Table 5). In all experiments, the input image size is 256x256.

We use the official VQGAN implementation[9] and model architectures for FacesHQ, where both encoder $f$ and decoder $g$ have 128 convolution channels. The spatial dimension of the latent representation is $16 \times 16$. See Table 7 and Table 8 in [21] for details. All experiments are run on eight V100 GPUs. For training the encoder and decoder, the learning rate is $4.5 \times 10^{-6}$, the batch

---

[9]https://github.com/CompVis/taming-transformers

| Dataset | Training | Validation | Total |
|---------|----------|------------|-------|
| CelebaHQ | 25k | 5k | 30k |
| FFHQ | 60k | 10k | 70k |
| FacesHQ | 85k | 15k | 100k |

Table 5: Number of images in the face datasets.

size is 8 on each GPU (total batch size 64), and the number of training epochs is 80. For training the transformer the learning rate is $2 \times 10^{-6}$ and the batch size is 12 on each GPU. The FIDs in Table 3 are obtained by selecting the best results from three independent runs, for both VQGAN and DAE-VQGAN.

To implement our 2-stage DAE-VQGAN, we consider two ways to realize the auxiliary decoder family $\mathcal{G}^A$ during the first stage of training the encoder. Similar to the settings in Section 6.1 and 6.2, we first utilize an extra auxiliary decoder $g_{aux}$ with half the number of channels as the actual decoder. Then, we experimented with decoder Dropout [73], as mentioned in Remark 4.3.

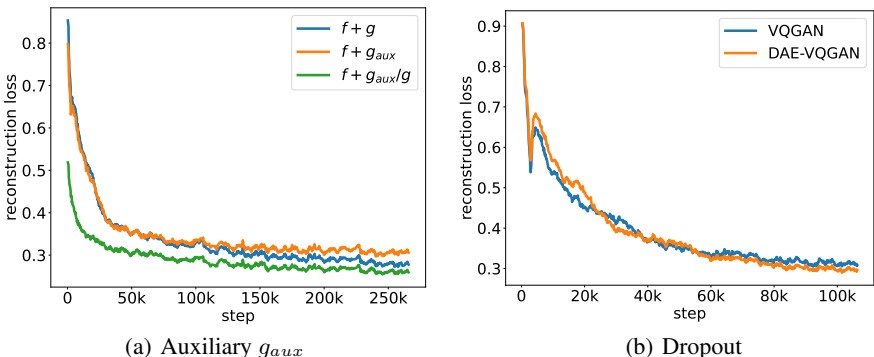

(a) Auxiliary $g_{aux}$  (b) Dropout

Figure 3: Training reconstruction loss trends for two implementations of DAE-VQGAN. (a) Halving the channels: while the reconstruction loss of our first stage ($f + g_{aux}$) converges to a higher value than the baseline ($f + g$), our second stage ($f + g_{aux}/g$) converges to a lower level with much faster convergence speed; (b) Dropout: the switch between the 2 stages of DAE occurs at halfway (40 epoch, around step 53k), where we can see that our method trails behind before the switch and surpasses the baseline after the switch and converges to a lower level of the reconstruction error.

**Halving the channels.** We use the same $f$ and $g$ architectures as the VQGAN baseline, and $g_{aux}$ is the same as $g$ except for the number of channels set to 64. To distinguish, we use $f + g$ to represent the models from the baseline VQGAN, and use $f + g_{aux}$ and $f + g_{aux}/g$ to denote the first stage and second stage of DAE-VQGAN, respectively. For the DAE training, we jointly train encoder $f$ and the auxiliary decoder $g_{aux} \in \mathcal{G}^A$ in the first stage (first 40 epochs). Then in the second stage (last 40 epochs), we replace $g_{aux}$ with $g$, and train $g$ from scratch with $f$ (and the trained codebook) fixed. For a more thorough investigation, we extended the run for both VQGAN and DAE-VQGAN to 200 epochs and the reconstruction error trends are shown in Figure 3(a), where we can see that our DAE-VQGAN achieves significantly better reconstruction. For the actual evaluation on the FacesHQ dataset, we stick to the 80 epoch training scheme for both the baseline and our DAE-VQGAN, with the switch point between the two stages at epoch 40.

**Dropout.** In the first stage, we implement the relatively weak auxiliary decoder by applying 2D Dropout (channel-wise) to the decoder. In particular, we replace the Dropout layer in each ResNet Block in the decoder of the original VQGAN with 2D Dropout and set ratio $p = 0.5$, and train the encoder and the dropped decoder jointly. Then in the second stage, we freeze the encoder and train the full decoder without Dropout. There are in total 80 training epochs for both the baseline and DAE-VQGAN (the first 40 epochs belong to the first DAE stage and the second DAE stage takes up the next 40 epochs). Figure 3(b) shows the trends of the reconstruction loss during training, where

we can see that our DAE-VQGAN achieves significantly better reconstruction as well, despite the fact that the encoder $f$ is only updated in the first stage.

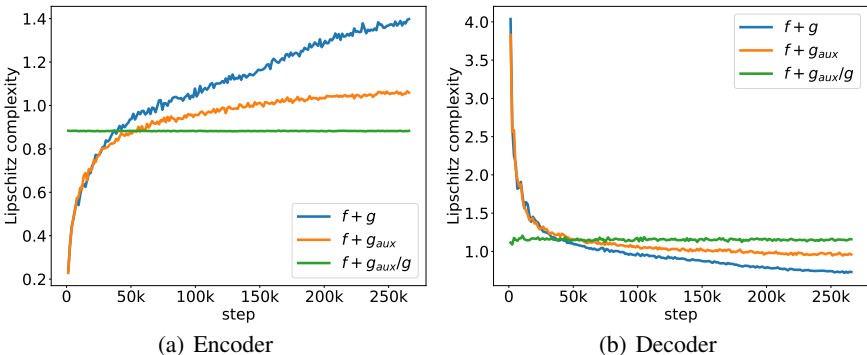

(a) Encoder           (b) Decoder

Figure 4: Lipschitz complexity of encoder and decoder during training. Our DAE second stage $f + g_{aux}/g$ utilizes the frozen encoder $f$ from the first stage $f + g_{aux}$ at epoch 40 (around step 53k, where the blue line and the orange line intersect). Since $g \circ f$ is trained to be the identity map, the trends for $C_{lip}(g)$ and $C_{lip}(f)$ are approximately reciprocal and the ideal values for them are 1. Compared with the baseline $f + g$ (blue line), our DAE stage one $f + g_{aux}$ (orange line) has a significantly lower $C_{lip}(f)$ that is closer to 1. Since the second stage $f + g_{aux}/g$ does not update the encoder, the green line stays flat.

## B.2 Complexity of encoder and decoder

One benefit of our DAE is the ability to better exploit the capacity/complexity of the encoder and decoder. To this end, we investigate the Lipschitz complexity $C_{Lip}$ of encoder $f$ and decoder $g$ defined as:

$$C_{Lip}(f) = \mathbf{E}_{x_1, x_2 \sim P_x} \left[ \frac{\|\tau(f(x_1)) - \tau(f(x_2))\|_2}{\|x_1 - x_2\|_2} \right], \tag{B.1}$$

$$C_{Lip}(g) = \mathbf{E}_{x_1, x_2 \sim P_x} \left[ \frac{\|g(\tau(f(x_1))) - g(\tau(f(x_2)))\|_2}{\|\tau(f(x_1)) - \tau(f(x_2))\|_2} \right].$$

Without loss of generality, we choose the DAE-VQGAN implemented by introducing an auxiliary $g_{aux}$ for demonstration. The conclusions for Dropout are similar. Figure 4 shows the Lipschitz complexity of the encoder and decoder during 200 epochs of training, where $f + g$ represents the baseline VQGAN, $f + g_{aux}$ represents the first stage of our method, and $f + g_{aux}/g$ represents the second stage of our method where encoder $f$ is initialized with weights from the 40-th epoch (around training step 53k) of the first stage training. Since $g \circ f$ is trained to be approximately the identity map, the trends of the encoder and decoder are the opposite. As stated in Section 3, the Lipschitz complexity should be ideally close to 1 for both the encoder and decoder. Compared with the baseline, our DAE-VQGAN produces a better latent distribution with significantly lower complexity overall (closer to 1) while achieving smaller reconstruction errors. We also compute a variant of $C_{Lip}$ where the $\ell_2$ norms of difference between $x_1$ and $x_2$, and that between their reconstructions, are replaced with perceptual similarity (LPIPS) [92]. This LPIPS complexity variant shows similar trends as $C_{Lip}$ (Figure 5).

## B.3 VQ codebook

We look into the codebook and investigate how different training strategies affect the behavior of the learned latent. For both VQGAN and DAE-VQGAN, the codebook size is 1024. For the codebook itself, we compute the pairwise cosine similarity among the learned codes and show the histogram of the similarity scores in Figure 6(a). For VQGAN, the cosine similarities are much more concentrated near 1 and -1, indicating that a large portion of learned codes may be highly similar. In comparison, codes learned by DAE-VQGAN are more scattered in the latent space and therefore can be more efficient. The expressiveness of our codebook is more powerful in approximating the ideal latent

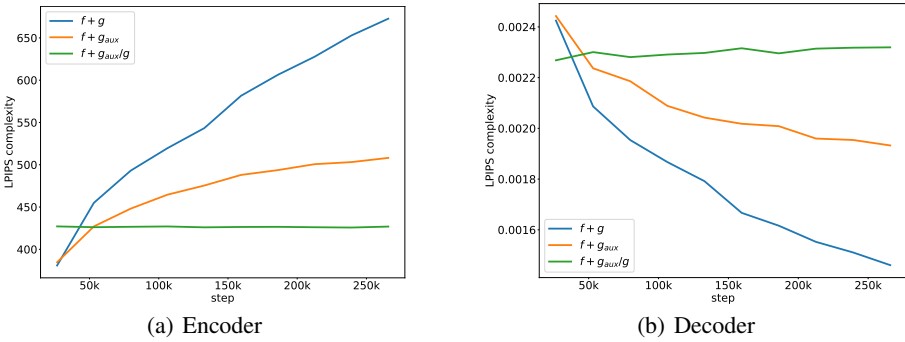

(a) Encoder            (b) Decoder

Figure 5: Lipschitz complexity with perceptual similarity (LPIPS complexity) of encoder and decoder. The overall trends are similar to those in $C_{lip}$ shown in Figure 4.

distribution than the original one. We further plot the eigenvalues of the cosine distance matrix for learned codes in Figure 6(b), where we can clearly see that the eigenvalue decay for the baseline is significantly faster, indicating more severe dimensional collapse.

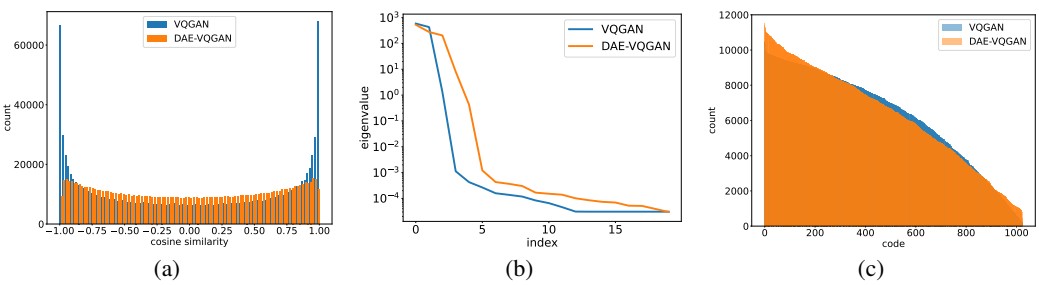

(a)            (b)            (c)

Figure 6: (a) Histogram of cosine similarity of codes. (b) Top 20 eigenvalues of the cosine distance matrix. (c) Number of appearances of codes in sorted order.

For the latent expressed using the codebook, we count and sort the number of appearances of codes over the training set of CelebAHQ (Figure 6(c)). We see that DAE-VQGAN can learn less redundant codes and hence their appearances are more evenly distributed. Interestingly, all codes are utilized in the reconstruction process for both VQGAN and DAE-VQGAN, perhaps due to the limited size of the codebook.

### B.4 Experiment details for DAE-DiT

We note that while the auxiliary decoder $g_{aux}$ approach presents clear trends of complexity changing, it requires manually changing the network between the two stages, and hence is less convenient to implement than the Dropout approach in practice. Therefore we adopt Dropout as the default approach in our DAE-DiT experiments.

Similar to DAE-VQGAN, in the first stage of DAE-DiT, we implement the relatively weak auxiliary decoder by applying Dropout[73] to $g$ with ratio $p = 0.2$, and train encoder $f$ and the dropped decoder jointly. Then in the second stage, we freeze encoder $f$ and train the full decoder $g$ without Dropout. We train on the OpenImages [46] dataset and adopt the KL-f8 model as in [68]. The encoder and decoder are trained with learning rate $4.5 \times 10^{-6}$ and batch size $64$.

For the diffusion training, we adjust the scale factor from the default 0.18215 to 0.8 such that the variance of features from the encoder is close to 1. Following the same setup as the official implementation, the EMA rate is 0.9999 and the classifier-free guidance scale is 4. The computation during both the training and the testing process was executed using FP16 precision except for the attention module for computational stability.

