# OpenReview forum: "Complexity Matters: Rethinking the Latent Space for Generative Modeling"
_NeurIPS.cc/2023/Conference — NeurIPS 2023 spotlight_

### Official Review · Reviewer_JCyg · 2023-06-16

**Soundness:** 3 good
**Presentation:** 3 good
**Contribution:** 4 excellent
**Rating:** 7
**Confidence:** 3

**Summary:**

This work investigates what constitutes a good latent space for generative models, and proposes a new training paradigm for generative models – DAE. Simply put, with DAE generative models are trained as Autoencoder in two stages. First, a relatively weak decoder is employed, whose purpose is to aid the encoder in learning meaningful representations. Second, the weak decoder is replaced with the “actual” decoder and training is continued. The second decoder is the one eventually evaluated as a generative model.


**Strengths:**

1. The paper is very well written, easy and interesting to follow.
2. The questions of what constitutes a good latent space and how is one constructed are often overlooked. Most works follow longstanding paradigms of using predetermined distributions (e.g., Gaussian) or whatever is learned from an autoencoder. Posing these questions and formulating the setting, on its own, is impactful.
3. The proposed method DAE is empirically proven effective, and since it is rather simple to implement, I conjecture it might have a strong impact on future generative modeling works.


**Weaknesses:**

While the mathematical formulation in the paper looks sound to me, I think it does not benefit the paper, and significant portions of it could be moved into the appendix. As the authors acknowledge in the Discussion section, the formulation“ serves mainly illustrative purposes” and is “not proven mathematically”. I don’t see it as an issue, as much of Generative Modeling research (and ML in general) is empiric in nature. The proposed method, DAE, could be introduced as an empirically-supported design, while some of the mathematical formulation could be described to serve as intuition (e.g., Theorem 4.1) . However, dedicating over three pages to it seems excessive to me. In my opinion, most readers would greatly benefit if the experiments in Appendix B were present in the main paper instead of some of the mathematical formulation.


**Questions:**

Could the authors please elaborate on “there are other algorithms to implement the
decoupled 2-stage training, and the evaluations in this work are not comprehensive”? Why would the authors not include such evaluations in the paper if they find them relevant?



**Limitations:**

Authors addressed limitations. One of them (non comprehensive evaluation) raises questions. I would appreciate a clarification on that.

---

> ### Author Rebuttal · Authors · 2023-08-10
>
> Thank you for the valuable comments and questions. Below we address them separately:
>
>
> ### 1. "The proposed method, DAE, could be introduced as an empirically-supported design, while some of the mathematical formulation could be described to serve as intuition ... In my opinion, most readers would greatly benefit if the experiments in Appendix B were present in the main paper instead of some of the mathematical formulation."
>
> > Thanks for the advice.
> > You are correct that DAE could be introduced as an empirically-supported design. However, we want to bring more insights into the latent distribution for generative modeling.
>
> > It is worth emphasizing that the DAE approach is an outcome and empirical verification of our investigation of the ideal latent distribution for generative modeling.
>
> > * Motivated by the GAN training objective, we first introduced $D^G$ to measure the closeness between the latent and the data in distribution. To the best of our knowledge, this work is the first to provide a characterization of the optimal latent distribution from the perspective of minimizing model complexity. Our proposed characterization has its own interest and may shed light on other applications as well.
> > * By utilizing an encoder, i.e., $P_z=P_{f(x)}$, we argue that minimizing $D^G(P_f(x), P_x)$ with respect to $f$ will give rise to the optimal latent. Notice that here we are considering matching in distribution, rather than sample-wise reconstruction. Therefore, we are advocating the use of adversarial training techniques as in VQGAN to train the encoder.
> > * We then identified the trade-off between the encoder and decoder, accompanied by a rigorous linear case analysis (Theorem 4.1). To address the trade-off, we proposed the decoupled approach DAE.
>
>
> > The experiments in Appendix B are certainly interesting. We will try to compress the mathematical formulation in the first part of the paper and put some of the results from Appendix B in the main text, to improve the reading experience.
>
>
> ### 2. Elaboration of "there are other algorithms to implement the decoupled 2-stage training, and the evaluations in this work are not comprehensive?"
>
> > The gist of our DAE modification is to make the decoder relatively weak in the first stage when training the encoder. By other algorithms, we meant that there are various ways to make a weak decoder. We only explored reducing the number of channels and 2D Dropout. Other candidates include reducing the number of layers, adding $l_2$ regularization, etc.
> Due to the computational budget, we only experiment with two common ways of regularization. The improvement is consistent in our evaluations and we believe other forms will also work.

---

> > ### Comment · Reviewer_JCyg · 2023-08-13
> >
> > Thanks for the rebuttal! It addresses the remarks I raised in my review.

---

### Official Review · Reviewer_14Q5 · 2023-06-20

**Soundness:** 3 good
**Presentation:** 3 good
**Contribution:** 2 fair
**Rating:** 6
**Confidence:** 3

**Summary:**

This paper presents an approach with theoretical analysis to explore a more suitable latent distribution for generation. For this purpose, this paper proposes a novel distance between the latent and target distributions and tries to minimize it to obtain the optimal data-dependent latent distribution. In practice, a two-stage training strategy called Decoupled Autoencoder (DAE) is introduced to leverage a superior latent distribution to improve the generative performance. The experiments on VQGAN and DiT show the effectiveness of the proposed method.

**Strengths:**

1.The paper is well-structured and easy to follow.

2.The methodology is supported by theoretical analysis.

**Weaknesses:**

1.The proposed methodology is somehow similar to AE-OT-GAN [1], which also learns a latent distribution via an autoencoder and then utilizes the learned latent to train GAN. The authors should explicitly highlight the advantages of their approach over [1].

[1] AE-OT-GAN: Training GANs from Data Specific Latent Distribution. ECCV 2020.

2.Some important related works are missed, particularly the line of works that explore improved latent sampling in GANs, e.g., [1] [2] [3] [4]. In special, AdvLatGAN [2] seeks to adjust and discover a more suitable latent distribution via adversarial learning in both the post-training sampling and the training, and it has already introduced the concept of optimal latent distribution noted $p_z^{op}$ in its theory.

[2] Improving Generative Adversarial Networks via Adversarial Learning in Latent Space. NeurIPS 2022.

[3] Discriminator optimal transport. NeurIPS 2019.

[4] Your gan is secretly an energy-based model and you should use discriminator-driven latent sampling. NeurIPS 2020.

3.The experimental results are insufficient. The comparison basically focuses on the performance gain beyond the vanilla backbones. Authors should compare the other efforts on improving the latent sampling/distribution, e.g., the previously mentioned works.

**Questions:**

1.As discussed in [2], the natural image distribution lie upon multiple disjoint manifolds, which potentially requires a discontinuous latent distribution (Prop. 3.2). Does the proposed method adhere to this requirement?

2.In Fig. 2 (a), actually it is hard to discern noticeable performance improvements in the comparison.

3.I am curious why this method can be applied to diffusion models, for which the Gaussian is the stable distribution for noise addition. If the latent distribution is altered, how does the mechanism of diffusion models still function?

Please also see the discussion of weaknesses. If the issues are resolved, I would be inclined to reassess and potentially raise my rating.

**Limitations:**

Please see above.

---

> ### Author Rebuttal · Authors · 2023-08-10
>
> Thanks for your time and effort in reviewing our work. There might be some misunderstanding and we have summarized and clarified the contributions of this work in the overall response. Hopefully, it can address some of your concerns.
>
> It is worth noting that although our motivation involves vanilla GAN models, our proposed methodology has an encoder-decoder structure that can be potentially adopted by a wider range of generative models that utilize a low-dimensional latent space.
>
>
> ### 1."The proposed methodology is somehow similar to AE-OT-GAN"
>
> > **The focus of our work and AE-OT-GAN is different**.
>
> > To clarify, we first summarize the procedure of AE-OT-GAN:
> > - (1) Train an autoencoder to embed data into the latent space (with distribution $\mu$) by minimizing the reconstruction loss;
> > - (2) Constructing $T$ with Semi-Discrete OT Map such that $T$ maps uniform $U[0, 1]^d$ to $\mu$;
> > - (3) GAN Training from $\mu$.
>
> > Our work shares the motivation of AE-OT-GAN but we investigate specifically the first step of AE-OT-GAN, i.e., the way to find a better latent.
> In comparison, the AE part in AE-OT-GAN follows standard procedures.
> In this sense, AE-OT-GAN can be seen as another example of a generative model that utilizes a latent distribution (induced by the encoder).
> Hence, our DAE modifications can also be applied to AE-OT/AE-OT-GAN. In the first step, we can utilize an auxiliary decoder to train a better autoencoder, potentially benefiting the afterward OT/GAN modeling.
>
> > We will add discussions to the AE-OT-GAN work in our revision.
>
>
> ###  2. "Some important related works are missed, particularly the line of works that explore improved latent sampling in GANs"
>
> > For the benefit of other viewers, let us first briefly summarize the related work [2-4].
> Both DOT [3] and DDLS [4] can be viewed as post-sampling latent space mining of GANs, by exploiting the trained discriminator (as discussed in the related work section of [4]).
> AdvLatGAN [2] further considered the GAN training process and proposed adding an implicit latent transform before the mapping function to improve the latent from its initial distribution. The implicit latent transform is trained by adversarial sample mining methods in a bi-level optimization fashion.
>
>
> > All aforementioned works can improve the performance of GAN models through latent space optimization.
> However, they do not characterize the **optimal latent distribution** and do not consider using an encoder network to parameterize/estimate it.
> As an encoder-decoder structure is popularly used in large-scale text-to-image generation models, such as stable diffusion, DiT, Muse, Parti, etc, we believe our investigation could bring new insights to them.
>
>
>
> ###  3."The experimental results are insufficient...Authors should compare the other efforts on improving the latent sampling/distribution..."
>
> > Our work focuses on improving the latent distribution of generative models with a low-dimensional latent space.
> The key practical contribution is the 2-stage DAE training of the autoencoders.
> For the baseline autoencoders, the popular training methods, vector-quantized or KL-regularized are both experimented in this work, to demonstrate the effectiveness of our new understanding.
>
> > Decoder-only GAN models are not our main target and [1-4] do not consider the autoencoder structure.
> Nevertheless, we have conducted DCGAN experiment on CIFAR-10 in Section 6.2 and the results are reported in Table 2.
>
> > Treating DCGAN as a baseline, we introduced 3 modifications.
> By DCGAN vs. DCGAN-SimCLR, we want to demonstrate that learning latent for generative modeling is different from standard self-supervised learning methods for discriminative tasks.
> By DCGAN vs VAEGAN, we want to demonstrate that a learned latent from data can be better than data-agnostic Gaussian, and VAEGAN can be further improved by our modified DAE-VAEGAN.
>
> > For larger-scale experiments, we considered VQ-GAN and DiT.
>
>
>
> ### Q1: "... natural image distribution lies upon multiple disjoint manifolds...Does the proposed method adhere to this requirement?"
>
> > Yes. If the data distribution lies upon multiple disjoint manifolds, the continuous encoder is capable of mapping them to multiple disjoint manifolds in the latent space. The resulting $P_{f(x)}$ can be multi-modal as well.
>
> > This is not a special property of our method. In fact, any AE-type methods can handle the challenge, as discussed in AE-OT. Our work made further improvements over the standard AE training and can result in a better latent distribution.
>
>
> ### Q2: "In Fig. 2 (a), actually it is hard to discern noticeable performance improvements in the comparison."
>
> > The figures are highly subjective and serve as a complement to the reported FID. Please mainly refer to the FIDs for a quantitative evaluation.
> We will move Fig 2(a) to the appendix and add more figures to showcase the difference.
>
> ### Q3: "I am curious why this method can be applied to diffusion models..."
>
>
> > In our work, the latent distribution $P_z$ refers to the **low-dimensional** distribution to be mapped to the data distribution $P_x$.
> The stable distribution of the forward process of diffusion models is sometimes referred to as the "latent distribution" as well. However, we do not call it the latent distribution in this work.
> We apologize for the confusion and we will add a remark to clarify the difference.
>
> > As we reiterated, our work focuses on finding a better latent distribution for generative modeling.
> Given the latent space, the generative modeling itself, whether it is autoregressive or diffusion, is not the focus of this work.
> For latent diffusion models, such as Stable-Diffusion and Diffusion Transformers, the diffusion modeling is on the latent space $P_z$, instead of $P_x$.
> Hence, our DAE modifications can be directly applied, not just to latent diffusion models, but potentially all generative models with an autoencoder structure.

---

> > ### Comment · Reviewer_ssMC · 2023-08-10
> > **Disjointed manifolds**
> >
> > For your answer to Q1, there is also a second part: the latent modeling has to be be compatible with multi-modal distributions. I think this is more crucial than the auto-encoder part which, being continuous, keeps the original topology of the natural images unchanged. But this is clearly not a limitation of the proposed method, it is rather a limitation of some of the existing generative models.

---

> > > ### Author Response · Authors · 2023-08-10
> > > **Author Response to Reviewer 2zRJ and 14Q5**
> > >
> > > We thank Reviewer ssMC for your valuable input in the discussion! We agree with your point that latent modeling should be compatible with multi-modal distributions.
> > >
> > > As pointed out by Reviewer 2zRJ, in standard VAE literature, if the decoder is too powerful will result in **posterior collapse**, that is the stochastic encoder maps data to the prior Gaussian distribution, which is not uni-modal and not informative.
> > >
> > > From this perspective, our DAE approach utilizes a relatively weak decoder when learning the latent with the aim to better retain the original manifolds in the latent space. As noted in our original response to Q1, if the data distribution is based on multiple disjoint manifolds, the encoder can map them to separate, disjoint manifolds in the latent space.

---

> > ### Comment · Reviewer_14Q5 · 2023-08-15
> > **Thanks for the rebuttal.**
> >
> > Thanks for the rebuttal. Most of the concerns have been addressed. However, I would still like to note some points regarding the related works.
> >
> > 1."AdvLatGAN [2] further considered the GAN training process and proposed adding an implicit latent transform before the mapping function to improve the latent from its initial distribution." "... they do not characterize the optimal latent distribution."
> >
> > This does not provide a concise summary of [2]. The more related part in [2] is Section 3.2.1, which optimizes the latent distribution to better match the data distribution. In [2]'s Section 3.2.1, the concept of optimal latent distribution (noted $p_z^{op}(G)$ in its theory) has already been included, and [2] claims its optimization could minimize the distance between the initial and optimal latent distribution. The motivation has already been explored to some extent by these previous works.
> >
> > 2."These methods do not consider using an encoder network to parameterize/estimate it."
> >
> > I understand the methodology difference. Nevertheless, the motivation for discovering an optimized latent distribution is consistent across these endeavors. This line of research should not be missed in the discussion.

---

> > > ### Author Response · Authors · 2023-08-16
> > > **Further clarification on related work**
> > >
> > >
> > > Thanks for the reply! We are glad that most of your concerns have been addressed.
> > >
> > > Below, we further address the additional questions/comments.
> > >
> > > ## Relationship to AdvLatGAN [2]
> > >
> > > We apologize that our summary of [2] was not comprehensive enough and some important discussions were not included in our initial response.
> > >
> > > First, we would like to emphasize the key differences between [2] and our work.
> > >
> > > To clarify, we stick to our notations and use lower-case $g$ to denote a generator and use upper-case $G$ to denote a family of generators.
> > >
> > > ### The **concept of optimality** is different
> > > * The so-called optimal latent $Z^{op}(g)$ in [2] is **conditioned on a fixed generator $g$** and the key property of $Z^{op}(g)$ is that $G[Z^{op}(g)]=x_r$ (Definition 3.1).
> > > In comparison, our optimal latent does not depend on any pre-defined generator and is characterized as the **closest in distribution to the data distribution $P_x$**.
> > > Although our definition involves capacity/complexity constraint on the family of generators $G$, the resulting optimal $P_z^*$ that minimizes $D^{G}(P_z, P_x)$ aims to directly reflect properties of $P_x$.
> > >
> > >
> > > * The definition $Z^{op}(g)$ contains **no granularity** and does not offer **quantitative measurement** of how close a latent is to the optimal one.
> > > For some cases, e.g., $g(z)\equiv 0$, no latent distribution can satisfy $G[Z^{op}(g)]=X_r$ and $Z^{op}(g)$ can be an empty set.
> > > In comparison, our distance-between-distributions characterization $D^G(P_z, P_x)$ provides a quantitative measurement for the goodness of a given latent $P_z$.
> > >
> > > Therefore, our concept of optimal latent distribution **is not included** in [2].
> > > Although both works discussed "optimal latent", **the definition and scope are fundamentally different**.
> > >
> > >
> > > ### The **parametrization of the optimal latent** is different.
> > >
> > > * In [2], the latent is modeled by a transformation $z^*$ from the original latent, e.g., standard Gaussian.
> > > In the proposed AdvLatGAN, there exist constraints on $z^*(z)$ that it cannot deviate too much from the identity map ($d(z_0, z)\le \epsilon$ in equation 5).
> > > In comparison, we proposed to model it by an encoder $f$ mapping from the original data, i.e., $P_z=P_{f(x)}$.
> > > We believe our parametrization is more flexible and may better capture the underlying structures of the data.
> > >
> > >
> > > ## Benefits of an Encoder
> > >
> > > As is iterated in both works, mapping uni-modal Gaussian to multi-modal distributions with disjoint supports is difficult and requires the mapping function to be highly complicated.
> > > Therefore, in [2], the ideal $z^*(z_0)$ (equation 5 in [2]) might also be highly complicated and hard to learn.
> > > We think this is an **inherent difficulty** when modeling the latent by transforming a pre-defined data-agnostic distribution.
> > > On the other hand, using an encoder can be more efficient in recovering the optimal latent. It is advocated in our work that the encoder-decoder structure should be used for generative modeling.
> > >
> > > On a side note, the sampling process of diffusion models also transforms a data-agnostic distribution to the target (in the same metric space). The inherent complexity is also an issue. However, the sampling process consists of multiple steps, and the complexity requirement can be easier to satisfy.
> > >
> > >
> > > Nonetheless, existing works on improving GAN, such as [2], are related to our work. You are totally correct that "the motivation for discovering an optimized latent distribution consistent across these endeavors". **We will add detailed discussions to them in the revision**.
> > >
> > > Thanks again for the questions! Hopefully, we have clarified the uniqueness of our work.

---

> > > > ### Comment · Reviewer_14Q5 · 2023-08-16
> > > > **Thanks for the clarification.**
> > > >
> > > > I appreciate the prompt efforts made to distinguish and highlight the distinctiveness of this work in comparison to previous research. The provided clarification is clear and logical to me. I raise my rating to 6.

---

### Official Review · Reviewer_2zRJ · 2023-07-05

**Soundness:** 3 good
**Presentation:** 2 fair
**Contribution:** 2 fair
**Rating:** 7
**Confidence:** 4

**Summary:**

The paper first proposes a new framework to analyze latent spaces in the context of generative models.
This framework takes inspiration from prior results about GANs, which allowed interpreting the min-max training objective as computing a distance between distributions to be minimized, to define a similar distance between the latent distribution and the data distribution.
From their analysis, they derive a simple two-step training for auto-encoders to learn better latents and reconstruction, in which they first train the encoder with a weak decoder to extract good latents, and then train a larger decoder to get better reconstructions.

They perform experiments in a simple toy case, and then with commonly used models such as GANs, VQGAN, and DiTs.

**Strengths:**

- The paper is easy to follow
- It proposes a novel view on latent codes, including an explanation of some different properties between SSL and generative latent codes, and a theoretical framework to describe why a powerful encoder/decoder pair can't learn a good latent code.
- The proposed practical solution is simple and their experiments show that it can improve performances in a variety of generative settings.

**Weaknesses:**

- While the fresh view on latent codes is interesting, it doesn't provide any theoretical guarantees.
As far as I understand, the main conclusion is that in order to obtain a good latent code the encoder and decoder should not be too powerful but is not able to give an indication about how the correct balance. Note that in the context of Variational Autoencoders, a similar conclusion had been reached before: that a too powerful decoder is a good explanation for the phenomenon called *posterior collapse* that describes a state in which the latent code is completely uninformative [1].
It is good that it is formally extended to other auto-encoders, but unsurprising.

- Because of this lack of quantification of optimal complexity, it is quite unclear whether the positive results obtained in VQGAN and DiT settings are actually related to the theoretical conclusions or not. It could very well be completely unrelated and just confirmation bias.

- Related work present different generative models with loose links to the proposed method, but does not discuss any study about the latent spaces of generative models. In addition to links with posterior collapse in VAEs [1], I would also be very interested to read what the authors have to say about sparsity and disentanglement properties of VAE [2], and PCA directions in GAN space [3] among other things.
Moreover, I have to point out that contrary to the paragraph in related works, Mask Autoencoders have been explored for their generative capabilities [4].


---
[1] Fixing a Broken ELBO. ICML 2018.
Alexander A. Alemi, Ben Poole, Ian Fischer, Joshua V. Dillon, Rif A. Saurous, Kevin Murphy.

[2] Challenging Common Assumptions in the Unsupervised Learning of Disentangled Representations. ICML 2019.
Francesco Locatello, Stefan Bauer, Mario Lucic, Gunnar Rätsch, Sylvain Gelly, Bernhard Schölkopf, Olivier Bachem

[3] GANSpace: Discovering Interpretable GAN Controls. NeurIPS 2020.
Erik Härkönen, Aaron Hertzmann, Jaakko Lehtinen, Sylvain Paris

[4] MaskGIT: Masked Generative Image Transformer. CVPR 2022.
Huiwen Chang, Han Zhang, Lu Jiang, Ce Liu, William T. Freeman

**Questions:**

One minor thing that was not clear to me is Remark 3.5. Could the authors explicit why scaling invariance would be a problem in this context?

In any case, despite its limitations, I still find the paper interesting for both its theoretical and practical contributions.
I trust the authors will fix the related work section, and my current rating anticipates that the authors will demonstrate their willingness to do so.

-----
Post-rebuttal update:
Having taken into consideration the rebuttal, discussions and comments from the other reviewers, I updated my score from WA to Accept.

**Limitations:**


The authors are very open regarding the limitations of their submission.

Potential negative societal impacts of generative models are not. Widely acknowledge ones include amplification of biases, potential misuse for propagating fake information, and concerns about lack of attribution and copyright infringements.

---

> ### Author Rebuttal · Authors · 2023-08-10
>
> Thanks for the valuable comments. Below we address them separately:
>
> ### 1. "While the fresh view on latent codes is interesting, it doesn't provide any theoretical guarantees."
>
> > Thanks for the question.
> You are correct that the main idea of our DAE approach is to balance the encoder and the decoder in different stages and our analysis did not offer precise algorithmic guidelines on how to achieve the optimal balance.
>
> > However, it is worth emphasizing that the DAE approach is an outcome and empirical verification of our investigation of the ideal latent distribution for generative modeling.
>
> > * Motivated by the GAN training objective, we first introduced $D^G$ to measure the closeness between the latent and the data in distribution. To the best of our knowledge, this work is the first to provide a characterization of the **optimal latent** distribution from the perspective of minimizing model complexity. Our proposed characterization has its own interest and may shed light on other applications as well.
> > * By utilizing an encoder, i.e., $P_z=P_{f(x)}$, we argue that minimizing $D^G(P_f(x), P_x)$ with respect to $f$ will give rise to the optimal latent. **Notice that here we are considering matching in distribution, rather than sample-wise reconstruction**. Therefore, we are advocating the use of adversarial training techniques as in VQGAN to train the encoder.
> > * We then identified the trade-off between the encoder and decoder, accompanied by a rigorous linear case analysis (Theorem 4.1). To address the trade-off, we proposed the decoupled approach DAE.
>
>
> > Making practical improvements to existing latent generative models is an important goal of this work. However, models such as VQGAN and DiT are too complicated for rigorous analysis. Instead, we draw inspiration from simple (linear) cases and verify the effectiveness on real scenarios mainly through experiments.
>
> ### 2. Relationships to VAE literature
>
> > This is a good point. As you pointed out, the fact that a decoder too powerful can result in posterior collapse is well known in VAE literature. Here we clarify the key differences to our work.
>
> > * Different target: This work focuses on the **ideal latent** distribution for generative modeling from the perspective of **minimizing complexity**. We consider mainly **deterministic** encoder and decoder where the decoder is the generator, and the encoder is what we used to parameterize the latent distribution. In our work, a good latent distribution is the primary target while in VAE, the latent distribution is often chosen in advance to be data-agnostic priors.
>
> > * Deterministic vs stochastic: To characterize the ideal latent distribution, we introduced $D^G(P_z, P_x)$, which is closely connected to GANs (Integral Probability Metrics or $f$-divergence). In comparison, VAEs usually maximize likelihood or mutual information $I$.
> It's worth noting that **while $I(x, z)$ can also serve as a "distance" between distributions in different dimensions, it will not work for our benefit** since we are considering deterministic encoders and decoders.
> In the deterministic encoder case, we have the conditional entropy $H(f(x)|x)=0$ and $I(x, f(x))=H(f(x))-H(f(x)|x)=H(f(x))$, where $H$ denotes entropy.
> That is, the mutual information would always be the entropy of the latent, hence uniformly distributed latent is always ideal.
>
> > In Fixing a Broken ELBO [1], the authors stated that obtaining a good ELBO is not enough for good representation learning. In practice, the authors proposed to target at a desired target rate $R^*$, such that a better balance between the informativeness of the latent and the reconstruction quality.
> As a result of the last bullet point, this approach cannot extend to the deterministic case.
>
> > In comparison, we focus on the complexity of the generator and we provided characterization of the optimal latent.
>
> > [1] also discussed realizability and their optimality is hinted to be the tightest achievable sandwiched bound within the parametric family. However, no formal statement is given.
>
> > Nevertheless, the VAE literature is related and we will add discussions to it in our revision.
>
>
> ### 3. "Related work presents different generative models with loose links to the proposed method, but does not discuss any study about the latent spaces of generative models."
>
> > Thanks for the suggested related works [1-4]. We will add a paragraph in the related work section to discuss this line of study in our revision.
>
> > Here we briefly discuss their relationships to our work.
>
>
> > Representation disentanglement is discussed in [2], which is not covered in our work and would be an interesting future direction to explore. [3] also follows this line of work and investigates the disentangled or interpretable latent variables or directions in GAN generation to control different aspects of the image, e.g., viewpoint, aging, etc.
>
>
> > In MaskGIT, the authors investigated how to better model the tokenized images, rather than finding better latent spaces to do masked modeling.
> Our insights about the optimal latent distribution and the DAE approach can also be beneficial in their settings.
>
>
> ### 4. "It is quite unclear whether the positive results obtained in VQGAN and DiT settings are actually related to the theoretical conclusions or not."
>
> > To empirically evaluate our DAE approach, we conducted a series of experiments from Gaussian mixture data, to DCGAN on CIFAR-10, to VQGAN and DiT on larger datasets. The improvements are consistent across different settings.
>
> > In practice, we provide a general rule-of-thumb on how to weaken the decoder in the first stage (cut the channels in half or add Dropout at strength 0.5).
> Such configurations work across different settings and we have not (and cannot afford to) done much fine-tuning, which supports the hypothesis that our theoretical insights are suitable and effective when applied to real-world model architectures.

---

> > ### Comment · Reviewer_2zRJ · 2023-08-10
> >
> > I thank the authors for their answers that adress the main points of my review:
> > - The rebuttal convincingly discuss their relation to other works.
> > - I am still not entirely convinced that the empirical results are a strong validation of the theory but the proposed links are reasonable. Also, both parts are valuable enough regardless.
> >
> > Ideally, I would still appreciate a clarification regarding my question on Remark 3.5.

---

> > > ### Author Response · Authors · 2023-08-10
> > > **Author Response to Reviewer 2zRJ: Explaination of Remark 3.5**
> > >
> > > Thanks for the feedback! We are glad that we have addressed most of your concerns.
> > >
> > > We are terribly sorry that we didn't explain Remark 3.5.  In the following, we first discuss the more general scaling problem, which is asked by Reviewer ssMC, and then specifically explain the arguments in Remark 3.5.
> > >
> > > The scaling problem might be easier to understand in the encoder-decoder case. Consider the optimal reconstruction case where $f=g^{-1}$ and let $C(\cdot)$ be the Lipschitz constant.
> > > For some $\lambda>1$, notice that $f_\lambda := \lambda\cdot f(x)$ and $g_\lambda:=g(x/\lambda)$ have the same reconstruction error, i.e., $g(f(x))=g_\lambda(f_\lambda(x))$ for any $\lambda\ne 0$. In this case, $C(f_\lambda)$ can be arbitrarily large by increasing $\lambda$.
> > > This scaling problem remains for the more general cases where $C(f)$ is not invariant to scaling.
> > >
> > > To address the scaling problem, we considered the generalized $D^{ae}(\cdot, \cdot)$, where both $C(f)$ and $C(g)$ are considered and we want their sum to be small.
> > > In this case, $C(f^*)=C(g^*)=1$ is the optimal choice since $C(f^*) + C(g^*) = C(f^*) + 1/C(f^*)\ge 2$ is minimized when $C(f^*)=1$.
> > >
> > > Now we go back to Remark 3.5. Let us use $P_f$ to represent $P_z$ and the generator has Lipschitz constant less than c, i.e., $g\in G_c$.  If we change $P_f$ to $P_{f_\lambda}$ for some $\lambda>1$, we can correspondingly consider $g_\lambda$, which will result in the same reconstructed distribution, i.e., $P_{g(f)}=P_{g_\lambda(f_\lambda)}$.
> > > Notice that $C(g_\lambda) < C(g)$ and is also in $G_c$. Therefore, $D^G(P_{f_\lambda}, P_x)$ is non-increasing with $\lambda$ and the optimal $P_{f_\lambda}$ might require $\lambda\to\infty$, which is ill-posed.
> > >
> > > For a more rigorous illustration, we refer to our Toy Example 1 in Appendix A.1.
> > >
> > > Please let us know whether we have sufficiently clarified Remark 3.5.

---

> > > > ### Comment · Reviewer_2zRJ · 2023-08-10
> > > >
> > > > Thank you for the very prompt and detailed clarification. I now see where it can be an issue.
> > > >
> > > > I consider all my comments adressed. I will take into account the new elements along with the updates from the other reviewers to formulate my final assessment, in any case before the end of the discussion period.

---

### Official Review · Reviewer_ssMC · 2023-07-06

**Soundness:** 3 good
**Presentation:** 3 good
**Contribution:** 3 good
**Rating:** 6
**Confidence:** 4

**Summary:**

This paper proposes an asymmetric training scheme for auto-encoders that double as image generator. Based on analytical insights that the decoder should have less capacity than the encoder for the encoder to capture correctly the data distribution, they propose a first training cycle where a strong encoder and a weak decoder are trained jointly. This produces a latent distribution able to better capture the data distribution. Then, in a second stage crucial for the end application, the encoder is frozen and a strong decoder is trained using the "good" latent distribution. Experiments are carried out on faces datasets with VQGAN and class conditional imagenet with a diffusion model and show the proposed training scheme is promising.

**Strengths:**

The strengths of this paper are :
- It is a simple method, yet effective. As such, it should be easy to reproduce. The improvements are maybe questionable (no error bars, test against the training set - as is traditional in this domain), but given that they are reported with widely different architecture and on different dataset, there is more confidence than usual that it actually works.
- The analysis part is really nice. The linear example in particular sheds some light as to why under a constrained budget, the complexity of the encoder should exceed that of the decoder to properly minimize the projected distance (divergence) to the data distribution.

**Weaknesses:**

The main weakness is that most of the paper is about the analytical part, which is a bit handwavy (acknowledged in the conclusion), and lacks a good structure. The reading could be improved by having a clear outline of the work such that one does not wonder where the text is headed to after each section.

**Questions:**

l.146 the link between the ideal $P_z$ and a low capacity generator is tenuous at best. There is no formal proof of that statement as far as I understand the paper. I can see a structural risk argument (ie, among all generator that equally minimize $D(p_x, p_z)$ chose the lowest complexity one), but it is more a principle and not a formal proof. Correct?
l. 190 $D(P_z, P_x)$ will be significantly larger in case of arbitrarily chosen $P_z$. This you don't know. If $P_x$ and $P_z$ have the same topology, then there exists a continuous mapping between the two that can be perfectly approximated by a sufficiently large neural network. In practice, maybe $D$ will be larger, but not necessarily in theory.
l. 198 I do not agree that $C(F) = C(G)$ is intuitive. For optimal reconstruction, we have $f = g^{-1}$ (assuming all that is required) and the Lipschitz constant of $g^{-1}$ can be arbitrarily large before that of $g$. For example, with $g(x) = x^2$ and $f(x) = \sqrt{x}$ on $(\epsilon, 1)$, the Lipschitz constant of $f$ can be made arbitrarily large. And vice versa. I think this depends whether it is easier (ie, the mapping has lower Lipschitz) to map from $x$ to $z$ or from $z$ to $x$. But what does it depend on? The size of $z$?
l. 312 this is a bit cheating. You should learn an encoder that inverse the (frozen) Gaussian space and evaluate on reconstruction for DCGAN.
l 354 Why is the Lipschitz of the encoder lower than that of the decoder for the proposed method? Isn't it completely contradictory to all the story of the paper?
A2. Fix notation errors.

**Limitations:**

The conclusion is a list of limitations that is very honest.

---

> ### Author Rebuttal · Authors · 2023-08-10
>
> Thanks for the valuable comments and questions. Below we address them separately:
>
>
> ### 1. "... most of the paper is about the analytical part ... lacks a good structure."
>
> > Thanks for the feedback.
> As is acknowledged, the analytical part mainly serves illustrative purposes and the proposed DAE is not proven mathematically.
> However, the DAE approach is an outcome and empirical verification of our investigation of the ideal latent distribution for generative modeling.
>
> > Making practical improvements to existing latent generative models is an important goal of this work. However, models such as VQGAN and DiT are too complicated for rigorous analysis. Instead, we draw inspiration from simple (linear) cases and verify the effectiveness on real scenarios through experiments. We conducted a series of experiments starting from toy Gaussian mixture data, to DCGAN on CIFAR-10, to VQGAN and DiT on larger datasets.
>
> > To improve readability, we summarized the contributions of this work in the overall response and we will incorporate them at the end of the Introduction section to provide an overview.
>
>
>
> ### 2. "l.146 The link between the ideal $P_z$ and a low capacity generator is tenuous ..."
>
>
> > The ideal $P_z^*$ is characterized as the one that minimizes $D^G(P_z,P_x)$.
> The link between its definition and the capacity of the generator **can be made mathematically sound with an extra assumption**.
>
> > First, recall that $G_c:=\{g\in G: C(g)\le c\}$ where $C(g)$ is defined as some complexity measurement. It is easy to see that for any $c_2 \ge c_1>0$, $G_{c_1} \subset G_{c_2}$. Therefore, $D^{G_{c_2}}(P_z, P_x) \le D^{G_{c_1}}(P_z, P_x)$.
>
> > Assumption 1: For any $P_z$ and $P_x$, $D^{G_c}(P_z, P_x)$ is continuous and monotonically decreasing with c.
>
> > Consider a generator family $G$ with bounded complexity and we target to have the generated distribution close to the data at level $\epsilon$ measured by $D$, i.e., $D(P_x, P_{g(z)})\le \epsilon$ for some $P_z$ and $g\in G$.
>
> > For a certain (non-degenerate) $P_z$, there exists $c>0$ such that $D^{G_c}(P_z, P_x) = \epsilon$, that is, by using this latent $P_z$, we need at least complexity $c$ to achieve the $\epsilon$ goal.
> If $P_z$ is not ideal, by definition we know that there exists $P_z^*$ that $D^{G_c}(P_z^*, P_x)=\epsilon^* < \epsilon$. By Assumption 1, we know there exists $c^\prime< c$ such that $D^{G_{c^\prime}}(P_z^*, P_x) = \epsilon$.
> This means that the goal can be achieved using a lower complexity generator if the latent is ideal.
>
> > Assumption 1 is not provable for general $D(\cdot, \cdot)$ and $C(\cdot)$. You are correct that this is more of a principle than a formal proof.
>
>
> ### 3. "l.190 $D(P_z,P_x)$ will be significantly larger in case of arbitrarily chosen $P_z$..."
>
> > You are right that if $P_x$ and $P_z$ have the same topology with a continuous mapping between the two, $D^{ae}(P_z,P_x)$ may not be significantly larger.
> > We will modify the statement to $D^{ae}(P_z,P_x)$ will be significantly larger in case of poorly chosen $P_z$ (e.g., uni-modal $P_z$ to multi-modal $P_x$) to make it more clear.
>
>
>
> ### 4. "l.198 I do not agree that $C(F)=C(G)$ is intuitive... l.354 Why is the Lipschitz of the encoder lower than that of the decoder for the proposed method?"
>
> > In the case of optimal reconstruction, we have $f=g^{-1}$ and the Lipschitz constant of $f$ can be arbitrarily large. This is related to our Remark 3.5 about scaling.
> Let $C(\cdot)$ be the Lipschitz constant.
> $k\cdot f(x)$ and $g(x/k)$ have the same reconstruction error.
>
> > To address the scaling problem, we considered the generalized $D^{ae}(\cdot, \cdot)$, where both $C(f)$ and $C(g)$ are considered and we want their sum to be small.
> In this case, $C(f^*)=C(g^*)=1$ is the optimal choice since $C(f^*) + C(g^*) = C(f^*) + 1/C(f^*)\ge 2$ is minimized when $C(f^*)=1$.
>
>
> > Going back to the line 198, it is worth emphasizing that, as stated in line 133, $C(\cdot)$ can be as specific as the Lipschitz constant, or as general as the size (width, depth, etc.) of the network.
> We will add remarks before line 198 to clarify that the complexity $C(\cdot)$ here is the latter general case.
> In practice, the encoder-decoder pair is often designed with symmetric architecture, with the same number of blocks and parameters, e.g., up-sampling vs. down-sampling, convolution vs. deconvolution, etc.
> The $C(F)=C(G)$ statement is not mathematically rigorous and we apologize for the confusion.
>
>
> > For the line 198 argument, we refer to **Figure 4** of the Appendix, where we can see that our DAE-modified encoder (orange line) is closer to one than the baseline (blue line). The same is true for the decoder.
> Given that the ideal case is that both are one, our DAE is effective at reducing the complexity.
>
> > As to whether the encoder or the decoder has a larger Lipschitz constant, we hypothesize that in the initial stage, it may closely relate to the mapping dimension. If $x_i$'s and $z_i$'s are of the same scale, $\|x\|_2/\|z\|_2\approx \sqrt{d/d_z}$. Hence, from high-dimensional $x$ to low-dimensional $z$, the Lipschitz constant might be smaller. This can be seen in the early stage of our Figure 4, where $C(f)\approx 0.2$ in the beginning.
>
> > However, after training, we do not know why the encoder has a larger Lipschitz constant.
>
>
>
> ### 5. "l.312 this is a bit cheating."
>
>
>
> > Treating DCGAN as a baseline, we introduced 3 modifications and the results are shown in Table 2.
> By DCGAN vs. DCGAN-SimCLR, we want to demonstrate that learning latent for generative modeling is different from standard self-supervised learning methods for discriminative tasks.
> By DCGAN vs VAEGAN, we want to demonstrate that a learned latent from data can be better than data-agnostic Gaussian, and VAEGAN can be further improved by our DAE approach.
>
>
> ### 6. "A2. Fix notation."
>
> > Thanks for pointing it out. We have fixed the singular values notation to $\lambda_i$ in line 576.

---

> > ### Comment · Reviewer_ssMC · 2023-08-10
> >
> > Thanks for the lengthy rebuttal! I think it covers well the remarks I had in my review.

---

> > > ### Author Response · Authors · 2023-08-10
> > > **Author Response to Reviewer ssMC**
> > >
> > > Thanks for the feedback! We are glad that we have addressed your concerns.

---

### Author Rebuttal · Authors · 2023-08-10

# Overall Response:

We thank all the reviewers for their time and efforts in reviewing our work. Before we address the questions and concerns of each reviewer, we would like to provide a summary of our work.

Our work aims to **characterize the ideal/optimal low-dimensional latent distribution for latent generative modeling**, from the perspective of minimizing the required **complexity** of the generator. The main contributions can be summarized as the following:

- Inspired by the training objective of GANs, we first propose a pseudo-distance $D^G$ between $P_x$ and $P_z$ and **characterize the optimal latent distribution $P_z^*$** by the one that is closest to $P_x$ in the terms of $D^G$. (Section 3.1)

- After characterizing $P_z^*$, we adopted the popular encoder parameterization of the latent distribution, i.e., $P_z = P_{f(x)}$ (Section 3.2). We then analyzed the interplay between the encoder and the decoder (generator) and identified the **trade-off between the quality/informativeness of the latent distribution and the capability of the decoder**. (Section 4.1)

- To address the trade-off, we proposed a **two-stage training scheme** called DAE that results in better latent spaces for more efficient generative modeling (Section 4.2). To verify our claims, we conducted experiments on various models such as GAN, VQGAN, and DiT, and achieved consistent improvement (Section 6).


Although our motivation involves vanilla GAN models, our proposed methodology has an encoder-decoder structure that can be potentially adopted by a wider range of generative models that utilize a low-dimensional latent space.

To the best of our knowledge, this work is the first to provide a characterization of the optimal latent distribution from the perspective of minimizing model complexity.
Our DAE approach is an outcome of our investigation along this line and an empirical verification of its practical effectiveness.
The proposed characterization has its own interest and may shed light on other applications as well.

---

### Decision · Program_Chairs · 2023-09-21

**Decision:**

Accept (spotlight)

**Comment:**

The paper makes a significant contribution to the field of generative modeling by providing a theoretical characterization of the optimal latent distribution for minimizing the required complexity of the generator. The authors propose a novel "distance" between the latent and data distributions, whose minimization coincides with that of the generator complexity. They then show that the minimizer of this distance can be parameterized by an encoder network, and they propose a two-stage training strategy called Decoupled Autoencoder (DAE) that can be used to learn this optimal latent distribution.

The authors conduct extensive experiments on various generative models, and they show that DAE consistently improves the quality of generated samples while also reducing the complexity of the generator. These results demonstrate the effectiveness of the proposed approach, and they make a significant contribution to the field of generative modeling.

The paper is well-written and easy to follow. The authors provide a clear and concise explanation of their approach, and they support their claims with strong experimental evidence. The paper is also well-referenced, and it makes a significant contribution to the existing literature.

I have no major concerns about the paper. The only minor issue is that the authors could provide more details about the implementation of DAE. However, this is a minor issue, and it does not detract from the overall quality of the paper.